# PLD3 affects axonal spheroids and network defects in Alzheimer's disease

Peng Yuan[1,2,10], Mengyang Zhang[1,3,4,6,10], Lei Tong[1,10], Thomas M. Morse[3], Robert A. McDougal[3,7], Hui Ding[1,8], Diane Chan[1,9], Yifei Cai[1] & Jaime Grutzendler[1,3,5]✉

The precise mechanisms that lead to cognitive decline in Alzheimer's disease are unknown. Here we identify amyloid-plaque-associated axonal spheroids as prominent contributors to neural network dysfunction. Using intravital calcium and voltage imaging, we show that a mouse model of Alzheimer's disease demonstrates severe disruption in long-range axonal connectivity. This disruption is caused by action-potential conduction blockades due to enlarging spheroids acting as electric current sinks in a size-dependent manner. Spheroid growth was associated with an age-dependent accumulation of large endolysosomal vesicles and was mechanistically linked with *Pld3*—a potential Alzheimer's-disease-associated risk gene[1] that encodes a lysosomal protein[2,3] that is highly enriched in axonal spheroids. Neuronal overexpression of *Pld3* led to endolysosomal vesicle accumulation and spheroid enlargement, which worsened axonal conduction blockades. By contrast, *Pld3* deletion reduced endolysosomal vesicle and spheroid size, leading to improved electrical conduction and neural network function. Thus, targeted modulation of endolysosomal biogenesis in neurons could potentially reverse axonal spheroid-induced neural circuit abnormalities in Alzheimer's disease, independent of amyloid removal.

Alzheimer's disease (AD) is a neurodegenerative condition that is characterized by widespread disruption in neural circuits and network connectivity[4]. The extracellular deposition of the β-amyloid (Aβ) peptide is thought to trigger a cascade of events, eventually leading to cognitive decline[5]. However, the cellular underpinnings linking Aβ deposition and neural network disruption are not well understood[6], limiting the rational design of new therapies. Extensive previous research has focused on mechanisms such as synapse loss and cell death as potential causes of neural dysfunction[7,8], and therapeutic efforts have mainly focused on strategies for extracellular amyloid removal[9]. However, an important and understudied pathological hallmark of AD is the markedly enlarged neuronal processes, traditionally termed dystrophic neurites, that are found around Aβ deposits[10–12]. These processes were previously shown to be all axonal rather than dendritic in origin[13,14] (Fig. 1a,b), so we therefore name them plaque-associated axonal spheroids (PAASs). Although various hypotheses regarding their development have been proposed over the years[15–19], these structures have not been a major focus of mechanistic investigation and their pathophysiological importance remains uncertain.

Here we undertook a multipronged approach to investigate PAASs using high-resolution structural imaging, time-lapse intravital calcium (Ca²⁺) and voltage imaging, as well as single-axon molecular manipulations and computational modelling. We found that hundreds of axons develop PAASs around each amyloid deposit and, rather than being associated with degenerative retracting neuronal processes, they are persistent and undergo dynamic changes in size over extended imaging intervals. PAASs markedly disrupt the propagation of action potentials (AP) by acting as electric current sinks in a spheroid-size-dependent manner, leading to abnormal long-range axonal connectivity and neuronal network dysfunction. We uncovered neuronal endolysosomal/multivesicular body (MVB) biogenesis as a critical determinant of PAAS size, and identified the neuronal lysosomal protein PLD3—a possible genetic risk factor for AD[1,20,21]—as a key mediator of endolysosomal abnormalities and PAAS enlargement. We further show that modulation of PLD3 levels can reverse axonal conduction abnormalities and restore neuronal network function in an AD mouse model. Together, our findings suggest a paradigm in which modulation of neuronal endolysosomal biogenesis can have a substantial effect on axonal electrical conduction and neural circuit function. Thus, targeting PAAS formation could be a strategy for ameliorating neural network abnormalities in AD.

## Functional impact of axonal spheroids

Axonal spheroids are found abundantly around individual amyloid plaques in both AD-like mice and human patients with AD (Figs. 1b

[1]Department of Neurology, Yale University, New Haven, CT, USA. [2]Department of Rehabilitation Medicine, Huashan Hospital, State Key Laboratory of Medical Neurobiology, Institute for Translational Brain Research, MOE Frontiers Center for Brain Science, Fudan University, Shanghai, China. [3]Department of Neuroscience, Yale University, New Haven, CT, USA. [4]Interdepartmental Neuroscience Program, Yale University, New Haven, CT, USA. [5]Wu Tsai Institute, Yale University, New Haven, CT, USA. [6]Present address: Department of Molecular and Cellular Physiology, Stanford University, Stanford, CA, USA. [7]Present address: Department of Biostatistics, Yale School of Public Health, Yale University, New Haven, CT, USA. [8]Present address: Department of Neurology, Xiangya Hospital, Central South University, Changsha, China. [9]Present address: Department of Neurology, Massachusetts General Hospital, Boston, MA, USA. [10]These authors contributed equally: Peng Yuan, Mengyang Zhang, Lei Tong. ✉e-mail: jaime.grutzendler@yale.edu

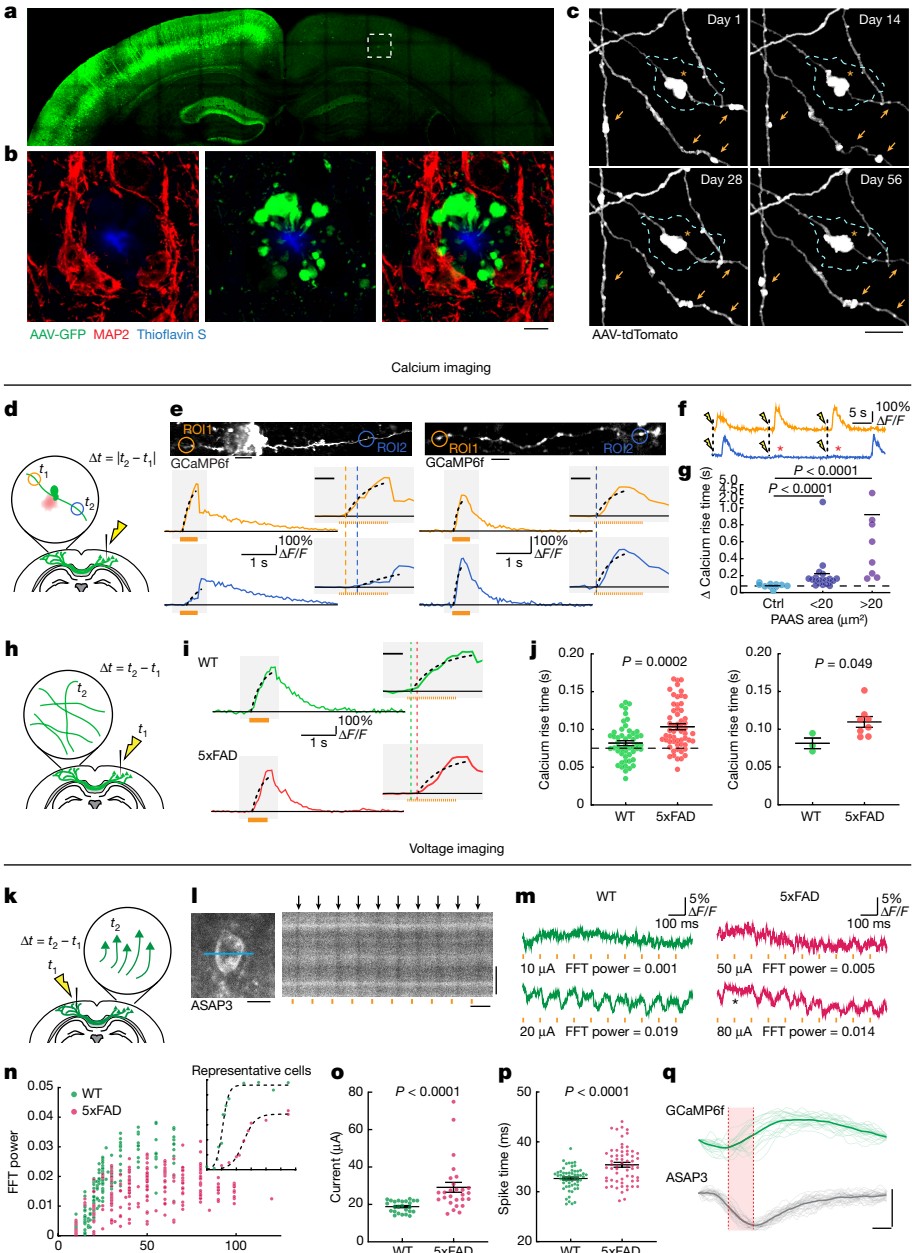

**Fig. 1 | Plaque-associated axonal spheroids block AP propagation and disrupt interhemispheric connectivity. a**, Confocal images of a mouse brain after unilateral injection of AAV2-GFP. **b**, Images from the region in **a** indicated by a dashed box showing axonal spheroids that can only come from transcallosal projecting axons (green) that are not associated with the dendritic marker MAP2 (red). Scale bar, 5 μm. **c**, In vivo two-photon time lapse of axonal spheroids near an amyloid plaque (cyan dashed lines), showing dynamic (arrows) and stable (asterisk) structures. Scale bar, 5 μm. **d**, Schematics of axonal calcium imaging at two sides of an axonal spheroid (green) near a plaque (red) after electrical stimulation of the contralateral hemisphere. **e**, Example of GCaMP6f-labelled axons with (left) and without (right) spheroids and traces of axonal calcium dynamics in ROIs at both sides of the plaque (orange and blue). The *y* axis indicates the Δ*F/F* of calcium transients. The orange bars show the 50 Hz stimulus pulse train. The insets show magnified plots (grey rectangles). The black dotted lines indicate exponential regressions of the rising phase. The orange and blue dashed lines show estimated calcium rise times. Scale bars, 10 μm (top images) and 200 ms (insets). **f**, Traces showing conduction blockade (asterisks) after stimulation (yellow flash icons). **g**, Differences in estimated calcium rise times at axonal segments on both sides of an individual spheroid. *n* = 10, *n* = 21 and *n* = 8 axons for the no spheroid, small spheroid and large spheroid groups, respectively; obtained from *n* = 14 mice. **h**, Stimulation and calcium-imaging strategies to measure long-range interhemispheric axonal conduction. **i**, Traces showing

of calcium dynamics in transcallosal axons imaged on the contralateral hemisphere. Scale bar, 200 ms (inset). **j**, The time interval between stimulation and the rise time presented by individual axons (left) or mice (right). *n* = 51 axons in *n* = 3 WT mice; *n* = 58 axons in *n* = 8 5xFAD mice. **k**, The strategy for axonal stimulation and two-photon voltage imaging of cell bodies to measure antidromic axonal conduction. **l**, Example of a voltage-sensor (ASAP3)-labelled cell body and the region of line scan (blue line) (left). Right, 1 kHz line scan kymograph after electrical stimulations (orange bars). The black arrows indicate APs. Scale bars, 10 μm (left and right vertical) and 100 ms (right horizontal). **m**, ASAP3 fluorescence traces. The orange bars indicate electrical stimulation. The electric current applied and the fast Fourier transform (FFT) power of 10 Hz are indicated below each trace. **n**, The probability of AP generation (FFT power) for each cell at a defined current. The inset shows examples of two individual cells at various current stimulations. **o**, The currents needed for 50% successful AP conduction for each cell (individual dots). *n* = 25 cells from *n* = 2 WT mice; *n* = 27 cells from *n* = 3 AD mice. **p**, The time interval between stimulation and AP spike time. *n* = 59 cells from *n* = 4 WT mice; *n* = 62 cells from *n* = 4 AD mice. **q**, The rise times (red shade) measured at the soma after stimulation of contralateral axons show similarity between GCaMP6f (green) or ASAP3 (grey). Scale bars, 10 ms (horizontal) and 1 arbitrary unit (AU) (vertical). Statistical analysis was performed using two-tailed Mann–Whitney *U*-tests (**g**, **j** (left), **o** and **p**) and a two-tailed paired *t*-test (**j** (right)). Data are mean ± s.e.m.

and 2i,l). On the basis of the average volume of individual PAASs and the total volume of the PAAS halo around plaques, we estimated that individual plaques can affect hundreds of axons on average (Extended Data Fig. 1a–c and Supplementary Discussion 1). Given the abundance of amyloid plaques in the AD brain, this suggests that substantial numbers of axons and their downstream interconnected neurons can be affected, highlighting the potential importance of PAASs as a mechanism of neural network dysfunction. In 5xFAD mice, time-lapse imaging of virally labelled axons around plaques revealed that PAASs can be very stable over intervals of up to months (Fig. 1c and Extended Data Fig. 1d–f). Although most PAASs increased in size over time, a substantial number decreased in volume or disappeared during this interval without the loss of the parent axon (Fig. 1c and Extended Data Fig. 1d–f), consistent with previous reports[11,15]. This supports the idea that PAASs are not a feature of degenerating axons but, rather, are stable structures that may affect neuronal circuits for extended intervals, while at the same time having the potential for reversibility.

To examine how PAASs might disrupt neuronal circuits, we computationally modelled how these spheroids could affect axonal electrical conduction. We found that the likelihood of disruption in the conduction of APs, for a particular axonal segment, markedly increased as a function of the total PAAS surface area (Extended Data Fig. 2). This in silico model predicted that PAASs behave as capacitors that function as electric current sinks for incoming APs and can therefore cause conduction blocks or prolonged delays (Extended Data Fig. 2, Supplementary Discussion 2 and Supplementary Video 1). To experimentally examine the electrical conduction properties of axons, we developed a strategy for measuring the propagation of APs in individual axons through $Ca^{2+}$ imaging in the live-mouse brain. We virally expressed the calcium sensor GCaMP6f through delivery of adeno-associated viral (AAV) vectors to one brain hemisphere (as in Fig. 1a) and performed $Ca^{2+}$ imaging of individual projection axons on the contralateral cortex (Fig. 1d). We measured AP propagation after electrically stimulating the ipsilateral hemisphere with trains of electrical pulses and compared the rise times of $Ca^{2+}$ transients at two regions of interest (ROIs) located on both sides of individual PAASs, along selected axonal segments (Fig. 1d). We found that the onset of the rise times was consistently delayed over intervals of tens to hundreds of milliseconds (Fig. 1e–g and Supplementary Video 2). Given that we used a series of pulses of electrical stimulation to induce trains of AP spikes, we concluded that the unusually long delays in the $Ca^{2+}$ rise times observed were due to conduction blocks of a substantial proportion of individual AP spikes, once they reached individual spheroids (Extended Data Fig. 2b). By contrast, a comparison of the rise times at two ROIs in axon segments without spheroids or at ROIs located on the same side of axonal segments adjacent to spheroids (Fig. 1e (right)) demonstrated no difference in the onset of rise times, strongly suggesting a deleterious effect of PAASs, rather than a broad spheroid-independent disruption in axonal electrical conduction. Furthermore, in addition to electrically evoked responses, we also imaged spontaneous $Ca^{2+}$ transients in individual axons and observed similar conduction abnormalities in segments with PAASs (Extended Data Fig. 3 and Supplementary Video 2). Regardless of whether the APs were the result of spontaneous or electrically induced neuronal activity, we observed that larger spheroids caused more severe conduction blocks (Fig. 1g and Extended Data Fig. 3d). These experimental observations were consistent with our computational modelling demonstrating that the size of individual PAASs is a critical determinant of the degree of axonal conduction defects (Extended Data Fig. 2f,f').

## Disruption of long-range connectivity

Given that we found marked abnormalities in local axonal conduction around plaques, we further investigated whether this was associated with more widespread defects in long-range cortical connectivity. Thus,

we developed a strategy for measuring interhemispheric conduction velocity through calcium imaging in live 5xFAD mice. To achieve this, we stereotaxically injected AAV9-Syn-GCaMP6f to label a homogeneous population of closely located cortical neurons in the somatosensory cortex, which assured comparable axonal distances to the imaged regions on the contralateral hemisphere across different mice. We then imaged contralateral projecting axons while electrically stimulating the ipsilateral GCaMP6f-labelled neurons (Fig. 1h). We found that the interhemispheric conduction velocities in wild-type (WT) mice were of a similar magnitude to those previously reported using slice electrophysiology recordings[22]. However, in 5xFAD mice, we found that $Ca^{2+}$ rise times in projecting axons were markedly delayed (Fig. 1i–j). This suggests that local AP conduction abnormalities caused by PAASs lead to a disruption in long-range axonal conduction. To further validate that our observations, as measured using calcium imaging, reflected actual AP delays or blockades, we implemented in vivo voltage imaging using the genetically encoded voltage sensor ASAP3[23]. We intracortically injected AAV2-Syn-ASAP3 in the same manner as we did for GCaMP6f (Fig. 1k). Given the relatively low signal-to-noise ratio when imaging with genetically encoded voltage sensors[23], it was not feasible to perform single-trial experiments to visualize AP propagation in individual axons. Instead, we stimulated axons in one hemisphere and recorded antidromic APs at neuronal cell bodies on the contralateral hemisphere (Fig. 1k,l), which markedly improved the signal-to-noise ratio. Using this strategy, we found that, in 5xFAD mice, there was a marked increase in the electric current required to induce the interhemispheric propagation of APs through single-trial stimulations (Fig. 1m–o), consistent with PAASs acting as current sinks. Furthermore, we also observed frequent delays in AP propagation when comparing 5xFAD and WT mice (Fig. 1p,q and Supplementary Discussion 3) in agreement with our $Ca^{2+}$ imaging experiments. Together, our imaging data as well as our computational modelling highlight the prevalence of AP conduction blocks resulting from spheroid pathology in AD-like mice.

Given the long interhemispheric distances, the probability of axons encountering amyloid plaques and developing spheroids is high. Although the density of amyloid plaques in humans is lower than that in mice, we hypothesize that the much greater axonal lengths in humans markedly increase the probability of adjacency to amyloid plaques and the likelihood of disruption in axonal connectivity. In support of this idea, through a quantitative analysis of post-mortem brains, we found a greater average number of axonal spheroids per amyloid plaque and larger PAAS size in individuals with moderate to severe AD, compared with those with mild cognitive impairment (MCI) (Extended Data Fig. 4a–c), consistent with previous literature[12,24]. Although this clinical–pathological correlation has various limitations, it suggests that both PAAS number and size could be important factors that determine the degree of neural circuit disruption and cognitive deficits in AD.

## Aberrant endolysosomes drive PAAS growth

High-resolution confocal imaging revealed that, as mice aged, there was a progressive accumulation of aberrantly enlarged lysosome-associated membrane protein 1 (LAMP1)-positive vesicles (ELPVs) within axonal spheroids (Fig. 2a,b). Transmission electron microscopy imaging revealed the accumulation of a heterogeneous population of vesicular organelles, including endosomes/MVBs, endolysosomes, amphisomes and autolysosomes (Fig. 2c and Extended Data Fig. 5), some of which could overlap with the ELPVs that we observed by optical imaging. This diversity potentially reflects distinct stages of organelle maturation through the lysosomal biogenesis and autophagic pathways[25,26]. There was a notable correlation between the presence of ELPVs and the overall size of individual spheroids (Fig. 2d). Furthermore, we also found that small PAASs were predominantly filled with vesicles that contained higher levels of the protease cathepsin D and were acidic—characteristic of more mature lysosomes (Fig. 2e–h and Extended Data

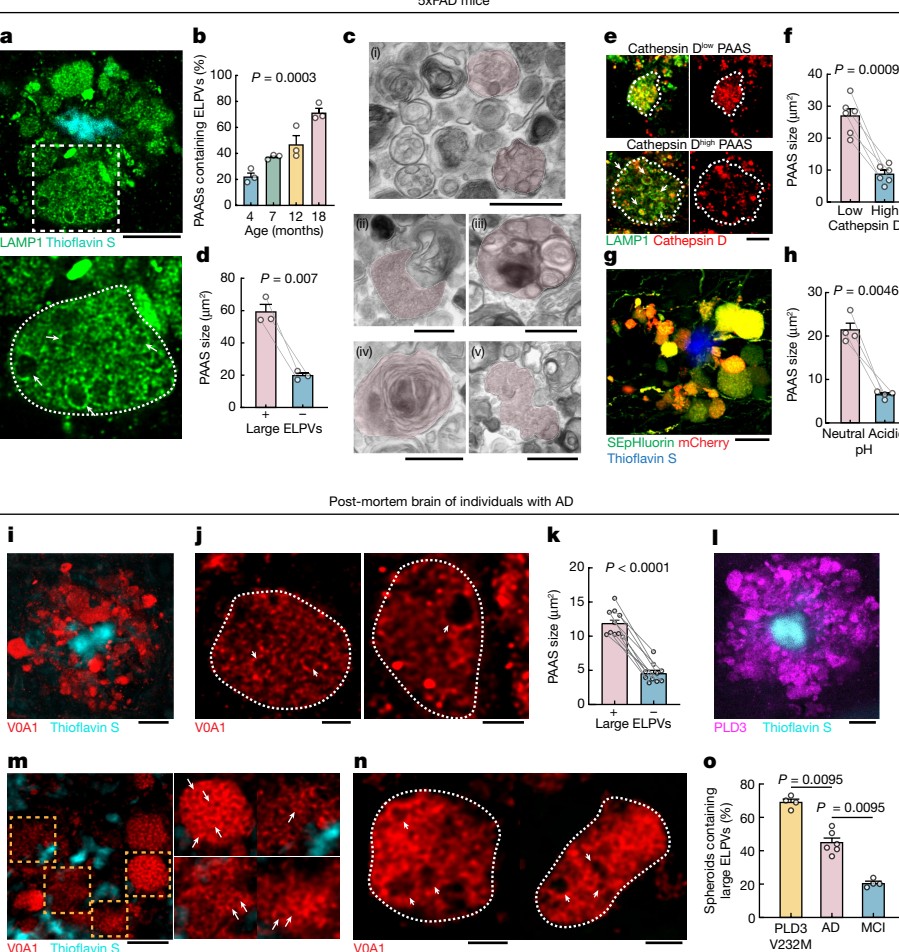

**Fig. 2 | Accumulation of abnormally ELPVs is associated with spheroid expansion and cognitive decline. a–h**, Analyses in 5xFAD mice. **a**, Confocal image of spheroids labelled by LAMP1 (green) around an amyloid plaque (cyan). Bottom, magnified image (dashed box in the top image) of a large spheroid (white dotted outlines). The arrows indicate ELPVs. Scale bar, 10 μm. **b**, ELPV occurrence within spheroids at different ages in 5xFAD mice. *n* = 3 mice for each group. **c**, Electron microscopy images of spheroids showing diverse endolysosomal and autophagic vesicles (red shade). MVBs (i); an MVB, possibly fusing with an autophagosome (ii); an organelle with ILVs (iii); an autophagosome (iv); and the fusion of multiple MVBs (v) are shown. Scale bars, 500 nm. **d**, Spheroid size with the presence or absence of ELPVs. *n* = 3 mice from each group. **e**, Confocal images of spheroids (white dotted lines) with high or low levels of cathepsin D. Scale bar, 5 μm. **f**, Spheroid size as a function of cathepsin D levels. *n* = 6 mice for each group. **g**, Confocal image of spheroids expressing the pH sensor SEpHluorin-mCherry. Scale bar, 10 μm. **h**, Spheroid size as a function of pH (red–green fluorescence ratio). *n* = 4 mice for each group. **i–o**, Analyses of post-mortem brains of individuals with AD. **i**, Confocal image showing human spheroids around an amyloid plaque labelled by V0A1 (red). Scale bar, 10 μm. **j**, Confocal imaging of individual V0A1-labelled spheroids (white dotted lines). The arrowheads indicate enlarged vesicles. Scale bars, 2 μm. **k**, Spheroid size with the presence or absence of enlarged V0A1-labelled vesicles. *n* = 10 human individuals from each group. **l**, PLD3 (magenta) enrichment within spheroids. Scale bar, 10 μm. **m,n**, Confocal images of spheroids labelled with V0A1 in the brain of a human with AD with the PLD3(V232M) variant. Right, images of individual spheroids indicated by the orange boxes. Enlarged vesicles are indicated by arrows. The arrowheads indicate enlarged vesicles. Scale bars, 5 μm (**m**) and 2 μm (**n**). **o**, Occurrence of enlarged vesicle in PLD3(V232M) variant, AD and MCI post-mortem human brain tissues. *n* = 4 individuals with AD with the PLD3 variant, *n* = 6 individuals with AD and *n* = 4 individuals with MCI. Statistical analysis was performed using a Kruskal–Wallis test (**b**), two-tailed paired *t*-tests (**d,f,h** and **k**) and two-tailed Mann–Whitney *U*-tests (**o**). In all of the graphs, each pair of dots represents the average measurement from spheroids in the same mouse and data are mean ± s.e.m.

Fig. 6f,g). By contrast, as PAASs increased in size, their overall acidification and cathepsin D levels declined (Fig. 2e–h and Extended Data Fig. 6f,g), consistent with the accumulation of ELPVs, which have not acquired sufficient lysosomal proteases and acidic pH[27]. Overall, this suggests that spheroid enlargement could be mechanistically linked to the accumulation of ELPVs.

Similar to mice, in post-mortem human brains, spheroids containing enlarged vesicles with low levels of cathepsin D (Fig. 2i,j and Extended Data Fig. 6a,b) were observed by immunolabelling of the lysosomal proton pump v-ATPase subunit V0A1 (Extended Data Fig. 6e,e′), and their presence was also associated with an overall larger spheroid size (Fig. 2k). Consistent with our observation that PAAS size is inversely

correlated with premortem cognitive function (Extended Data Fig. 4b), we found a similar correlation between premortem cognition, the abundance of large ELPVs and low levels of cathepsin D within PAASs (Fig. 2o and Extended Data Fig. 6c,d). Together, our mouse and human data support a hypothesis in which the accumulation of ELPVs may drive the enlargement of axonal spheroids, leading to the disruption of axonal conduction and ultimately cognitive dysfunction.

## Role of PLD3 in spheroid enlargement

We next investigated the potential mechanisms of ELPV accumulation within axonal spheroids. PLD3 is a lysosomal protein[2,3,28] that is

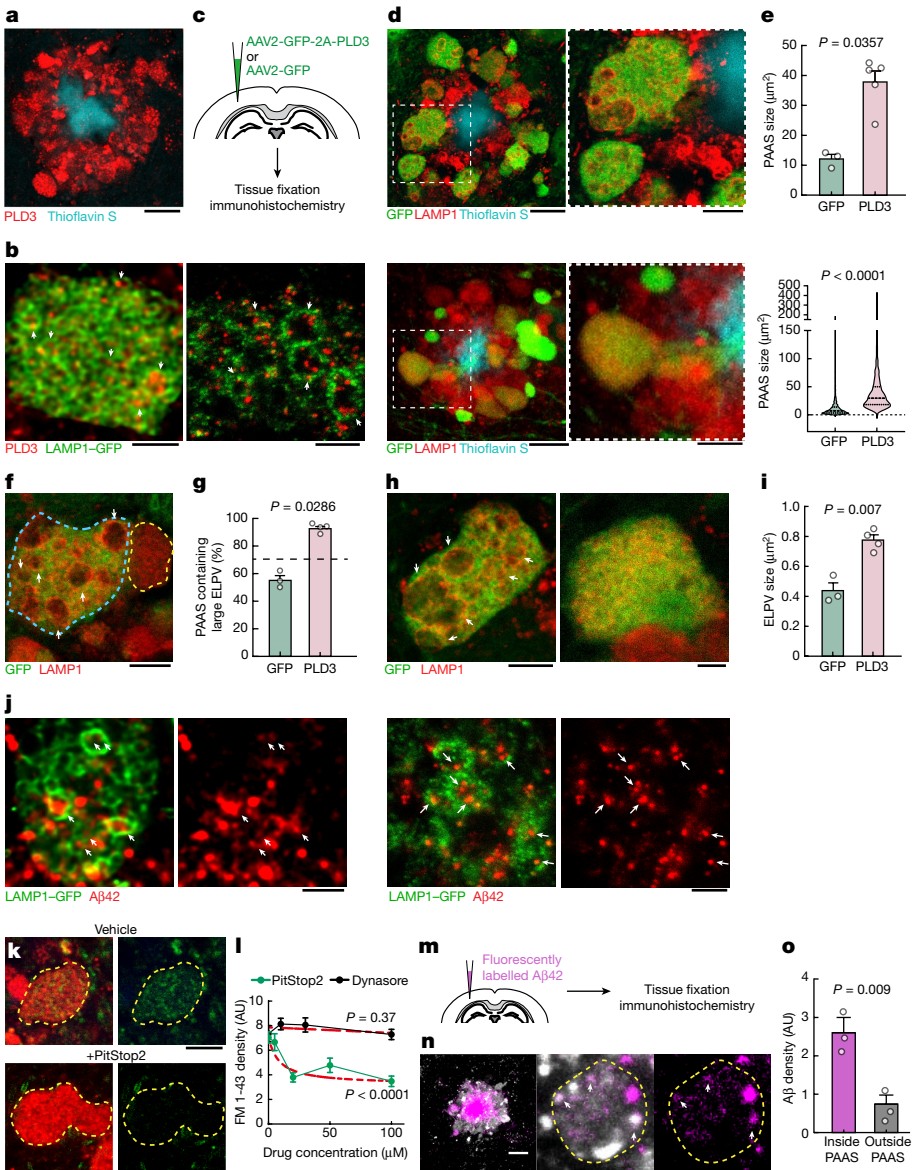

**Fig. 3 | PLD3 mediates endolysosomal vesicle enlargement and spheroid expansion. a**, Confocal image showing the enrichment of PLD3 in spheroids in 5xFAD mice. Scale bar, 10 μm. **b**, Confocal (left) and expansion microscopy (right) images of PLD3 immunofluorescence (red) and LAMP1–GFP (green) in spheroids. The arrows indicate PLD3 puncta within ELPVs. Scale bars, 2 μm (left) and 5 μm (right). **c**, Schematics of AAV-mediated PLD3 overexpression. **d**, Confocal images of spheroids after PLD3 (top) or control GFP (bottom) overexpression in 10-month-old 5xFAD mice. Right, magnified image of the area indicated by a white dashed box. Scale bars, 10 μm (left) and 5 μm (right). **e**, Spheroid size in 10-month-old 5xFAD mice with PLD3 or GFP overexpression, quantified by individual mice (top) or spheroids (bottom). *n* = 3 and *n* = 5 mice for the GFP and PLD3 groups, respectively. Each dot represents average measurements from 350–600 individual PAASs. The violin plots show the distributions of around 1,000–1,500 individual spheroids from each group. **f**, Confocal images of adjacent spheroids with (blue dashed line) and without (yellow dashed line) PLD3 overexpression. The arrows indicate ELPVs. Scale bar, 5 μm. **g**, ELPV occurrence in spheroids of 10-month-old 5xFAD mice with PLD3 or GFP overexpression. *n* = 3 and *n* = 4 mice for the GFP and PLD3 groups, respectively. Each dot represents the average measurement from 150 to 200 individual spheroids. **h**, Confocal images of spheroids in 5xFAD mice with PLD3 (left) or GFP (right) overexpression. The arrows indicate ELPVs. Scale bars, 5 μm

(left) and 2 μm (right). **i**, ELPV size in spheroids of 10-month-old 5xFAD mice with PLD3 or GFP overexpression. *n* = 3 and *n* = 4 mice for the GFP and PLD3 groups, respectively. Each dot represents the average measurement from 500–1,000 ELPVs. **j**, Confocal (left two images) and expansion microscopy (right two images) images of Aβ42 immunofluorescence (red) and LAMP1–GFP (green) within spheroids. The arrows indicate Aβ42 puncta contained within ELPVs. Scale bars, 2 μm (left two images) and 5 μm (right two images). **k**, Confocal images showing incorporation of FM 1-43 dye into spheroids in cultured brain slices after treatment with vehicle or PitStop2. Scale bars, 10 μm. **l**, Quantification of FM 1-43 incorporation into PAASs after PitStop or Dynasore treatment. *n* = 20 spheroids for each group. The red dashed lines show regression to a sigmoid inhibition curve. **m**, In vivo assay to measure Aβ endocytosis into spheroids after intraparenchymal brain microinjections of fluorescently labelled Aβ42 peptide. **n**, Confocal images of fluorescently tagged Aβ42 (magenta) incorporated into spheroids (white). The arrows indicate Aβ42 puncta. Scale bars, 10 μm (left) and 5 μm (middle and right). **o**, Quantification of Aβ42 incorporation into spheroids. *n* = 3 mice, each with average measurements from *n* = 10 fields of view. Statistical analysis was performed using two-tailed Mann–Whitney *U*-tests (**e** and **g**) and two-tailed Welch's *t*-tests (**i** and **o**), and *F*-tests were used to compare the fitted top and bottom parameters for each group in **l**. Data are mean ± s.e.m.

of potential interest because it strongly accumulates in axonal spheroids in both humans and mice[28,29] (Figs. 2l and 3a) and its expression is not detectable in other cell types such as microglia and astrocytes (Extended Data Fig. 7), despite mRNA presence in glial cells[30]. Moreover, *PLD3* genetic variants may increase the risk of AD, although this remains a controversial topic[1,20,21]. We found that there was an overall increased abundance of ELPVs in axonal spheroids of patients with AD who have the *PLD3* variant V232M[1] (Fig. 2m–o), suggesting a role of this variant in aberrant axonal endolysosomal function.

During endolysosomal maturation, the limiting membrane of late endosomes invaginates to form intraluminal vesicles (ILVs) and become MVBs that fuse with lysosomes[25]. Furthermore, autophagic degradation also requires the fusion of autophagosomes and MVBs, forming amphisomes that later fuse with lysosomes[31]. Thus, MVBs are crucial intermediate organelles that connect with various components of endocytosis, autophagy and lysosomal degradation[26,31] (Extended Data Fig. 5). PLD3 is unique among lysosomal-resident proteins in that it is sorted to the ILVs of MVBs[3], in contrast to most lysosomal-resident proteins, which are sorted to the limiting membranes of MVBs[32]. Indeed, immunofluorescence confocal and expansion microscopy imaging of axonal spheroids showed an accumulation of a punctate PLD3 signal within the lumen of LAMP1-positive vesicular structures (Fig. 3b), similar to previous immunogold electron microscopy analysis of cultured cells[3]. Together, these observations raise the possibility that PLD3 may have a role in MVB biogenesis, thereby affecting various organelles that interact with MVBs. This may lead to the accumulation of ELPVs, driving the expansion of axonal spheroids.

To further understand the role of PLD3 in the evolution of axonal spheroid pathology, we implemented in vivo AAV2-mediated overexpression of PLD3 in neurons of 5xFAD mice (Fig. 3c). We observed that spheroids in PLD3-overexpressing axons were markedly larger than those expressing only green fluorescent protein (GFP) (Fig. 3d,e and Extended Data Fig. 8a,b′). Notably, confocal microscopy of individual spheroids revealed an increase in the number of large ELPVs, even beyond what is seen in older 5xFAD mice (Fig. 3f,g and Extended Data Fig. 8c,d). Furthermore, there was a marked increase in ELPV size within PLD3-overexpressing spheroids compared with the GFP-expressing controls (Fig. 3h,i). This manipulation was not associated with changes in amyloid plaque number or size (Extended Data Fig. 8e,f), suggesting that PLD3 overexpression does not affect the processing of amyloid precursor protein (APP), in agreement with previous in vivo and in vitro experiments[2].

The effect of PLD3 overexpression on the accumulation of large LAMP1-positive vesicles was predominantly seen in axonal spheroids around plaques rather than in neuronal cell bodies (Extended Data Fig. 8g,h). This suggests that extracellular Aβ deposits are critical for PLD3-induced endolysosomal abnormalities in axonal spheroids. Indeed, using an antibody that specifically recognizes the Aβ42 peptide, we observed Aβ accumulation within large ELPVs in spheroids (Fig. 3j). The source of Aβ42 within ELPVs is likely to be the endocytosis of oligomeric peptides from adjacent amyloid plaques. This is supported by our data showing that spheroids are sites of active endocytosis (Fig. 3k,l), and that administration of fluorescently labelled Aβ42 to 5xFAD mice leads to robust uptake into vesicular structures within spheroids (Fig. 3m–o). Previous reports have demonstrated the formation of ELPVs in vitro after Aβ administration[33]. Furthermore, our data with PLD3 overexpression in WT mice also led to occasional formation of small axonal swellings with ELPVs (Extended Data Fig. 8i), suggesting that excessive PLD3 by itself can have detrimental effects, independent of amyloidosis. Together, these data suggest that the accumulation of PLD3, observed in both mice and humans, is mechanistically linked with endolysosomal abnormalities and the subsequent enlargement of axonal spheroids. This may be compounded by the concurrent effect of PLD3 and Aβ accumulation within the same subcellular compartments.

## *Pld3* deletion restores axonal conduction

To assess whether reducing PLD3 levels would ameliorate axonal spheroid pathology, we deleted *Pld3* in neurons using AAV2-mediated CRISPR–Cas9 gene editing in 5xFAD mice, using either of two single guide RNAs (sgRNAs), targeting different *Pld3* exons (Fig. 4a and Extended Data Fig. 9). We found that treatment with both sgRNAs led to a marked decrease in the abundance of large ELPVs (Fig. 4b,c and Extended Data Fig. 10d), which was associated with an overall reduction in PAAS size, regardless of whether the treatment was initiated at 3 or 7 months of age in 5xFAD mice (Fig. 4d–e and Extended Data Fig. 10a–c′). These data demonstrate that *Pld3* deletion at early or later stages of amyloid deposition can decrease ELPV accumulation in axons, leading to a marked reduction in spheroid growth, without any changes in amyloid plaque number or size (Extended Data Fig. 10e,f).

To test whether deletion of *Pld3* in neurons and the consequent reduction in spheroid size had a beneficial effect on axonal conduction, we co-infected neurons with AAV2-U6-sgRNA(Pld3)-CAG-Tomato-P2A-Cre to delete *Pld3* and AAV9-Syn-GCamP6f to implement our Ca²⁺-imaging approach for measuring interhemispheric axonal conduction (Fig. 4f). This coinfection strategy enabled us to image adjacent axons with or without *Pld3* deletion within the same mouse and compare the Ca²⁺ rise times in the contralateral cortex after electrical stimulation of the ipsilateral hemisphere (Fig. 4f). Using the two sgRNAs, we found that axons with *Pld3* deletion had a marked improvement in the propagation of APs (Fig. 4g,h and Extended Data Fig. 11a,a′,c,d′) that approached what we observed in control non-AD mice (Fig. 1j). By contrast, PLD3 overexpression had the opposite effect, with worsening of AP propagation observed in axons with increased PLD3 compared with adjacent controls (Fig. 4i–j and Extended Data Fig. 11b,b′). Together, these data demonstrate that PLD3 reduction can ameliorate endolysosomal abnormalities in axons near plaques, leading to reduced spheroid size and restored axonal conduction properties.

## PAAS reduction improves network function

To examine the effect of reversing spheroid-associated axonal conduction defects on neural circuit function, we focused on the basal forebrain nucleus of Meynert, which is a major source of cholinergic neurotransmission with widespread axonal projections that exert complex neuromodulatory effects on cortical neurons[34]. Furthermore, cholinergic networks are critical for normal cognitive function and are affected in the early stages of AD[35]. We therefore investigated the potential network effects of spheroid-induced conduction deficits and their potential reversibility by PLD3 modulation.

To achieve this, we injected AAV2-U6-sgRNA(Pld3)-CAG-Tomato-P2A-Cre to delete *Pld3* in neurons of the basal forebrain in 7-month-old 5xFAD mice. To examine the effect of improved basal forebrain neurotransmission, we imaged spontaneous Ca²⁺ transients during awake resting sessions in neurons of layer 2/3 of the somatosensory cortex, previously infected with AAV9-Syn-GCamP6f (Fig. 5a–c and Supplementary Video 3). These neurons were within the immediate vicinity of projecting basal forebrain axons (Fig. 5a (right)). We observed a higher proportion of hyperactive neurons in 5xFAD mice (Fig. 5d), consistent with previous reports[36]. Moreover, we also found that 5xFAD mice showed increased correlated activity in neurons that were in proximity to each other (Fig. 5e) and displayed activity patterns with higher spatiotemporal similarities (Fig. 5f,g). These aberrant patterns of activity are relevant because they are predicted to markedly disrupt efficient information encoding[37]. *Pld3* deletion in basal forebrain neurons led to a reduction in such aberrant cortical neuronal activity patterns to levels that were similar to WT controls (Fig. 5d–g). These data demonstrate that reversing spheroid-induced AP blockades in basal forebrain projection axons, which potentially improves neuromodulatory neurotransmission, can ameliorate the aberrant patterns of activity of downstream cortical neurons.

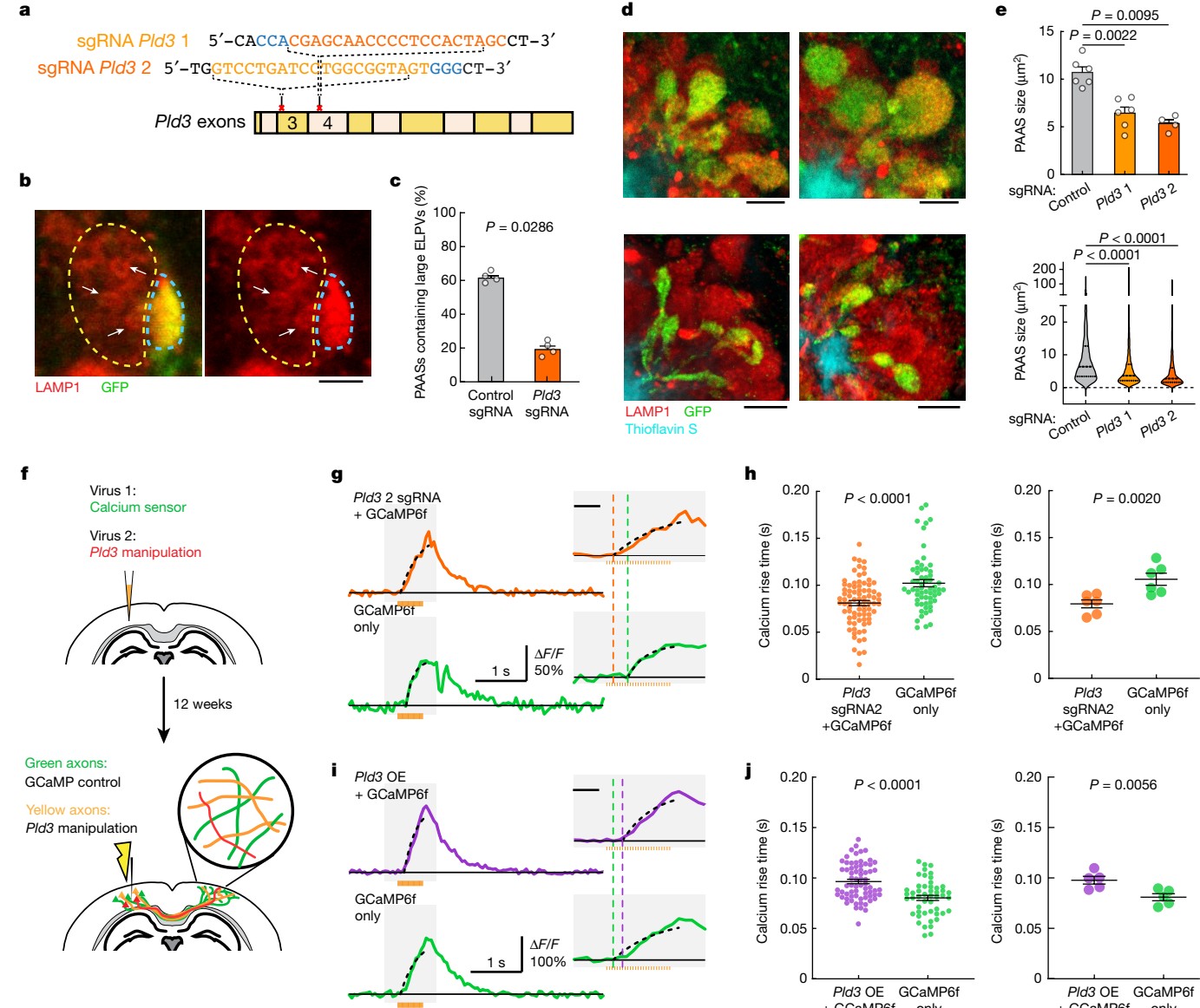

**Fig. 4 | CRISPR–Cas9-mediated *Pld3* deletion reduces spheroid size and improves axonal conduction. a**, The design of two guide RNAs targeting the *Pld3* gene. **b**, Confocal images of adjacent spheroids with (blue dashed line) and without (yellow dashed line) *Pld3* deletion. The arrows indicate ELPVs. Scale bar, 2 μm. **c**, ELPV occurrence in *Pld3*-deleted and control PAASs in 10-month-old 5xFAD/LSL-Cas9 mice. *n* = 4 mice for each group. Each dot is the average measurement of 150 to 250 individual spheroids. **d**, Confocal images of spheroids expressing control sgRNA (top) or *Pld3*-targeting sgRNA (bottom) in 10-month-old mice, showing infected (GFP⁺) and uninfected (red) spheroids around an amyloid plaque (cyan). Scale bars, 5 μm. **e**, Spheroid sizes in 10-month-old mice with or without *Pld3* deletion, presented by individual mice (top) or individual spheroids (bottom). *n* = 6, *n* = 6 and *n* = 4 mice for control sgRNA, *Pld3* sgRNA 1 and *Pld3* sgRNA 2, respectively; each dot represents the average measurements of 350–600 individual spheroids (top). The violin plots show the distributions of around 1,200–2,600 individual PAASs from each

group (bottom). **f**, Schematics of calcium imaging to measure conduction in contralateral axons with (yellow) or without (green) *Pld3* manipulation. **g**,**i**, Example traces of calcium dynamics in contralateral axons after *Pld3* deletion with sgRNA 2 (**g**) or overexpression (**i**). The orange bars show the 50 Hz spike train for stimulation. The insets show magnified plots of the calcium transients (grey rectangles). The black dotted lines indicate exponential regressions of the rising phase and the coloured vertical dashed lines show the estimated spike times. For the insets in **g** and **i**, scale bars, 200 ms (insets). **h**,**j**, Spike times in *Pld3*-deletion (**h**), *Pld3*-overexpression (OE) (**j**) or control axons, shown by either individual axons (left) or mice (right). *n* = 80 manipulated and *n* = 61 control axons from *n* = 6 mice with *Pld3* deletion; *n* = 69 manipulated and *n* = 49 control axons from *n* = 5 mice with *Pld3* overexpression. Statistical analysis was performed using two-tailed Mann–Whitney *U*-tests (**c**,**e**,**h** (left) and **j** (left)) and two-tailed paired *t*-tests (**h** (right) and **j** (right)). Data are mean ± s.e.m.

## Discussion

Here we show that hundreds of axons around each amyloid plaque develop spheroids and, rather than being retraction bulbs from degenerating axons, these structures are stable for extended periods of time and could therefore have an ongoing detrimental effect on neuronal connectivity. Given the similarity in the morphology, organelle and biochemical content of PAASs in mice and humans, it is probable that,

in humans, these are also stable structures that could disrupt neural circuits for extended intervals. To better understand the effect of PAASs on axonal function, we implemented in vivo Ca²⁺ and voltage imaging in individual cortical axons and cell bodies. Both Ca²⁺ and voltage imaging revealed that a substantial proportion of axons in 5xFAD mice had disrupted AP conduction and an overall increase in the threshold for AP propagation manifested by conduction blockades. This was due to the presence of axonal spheroids and was shown to be correlated with

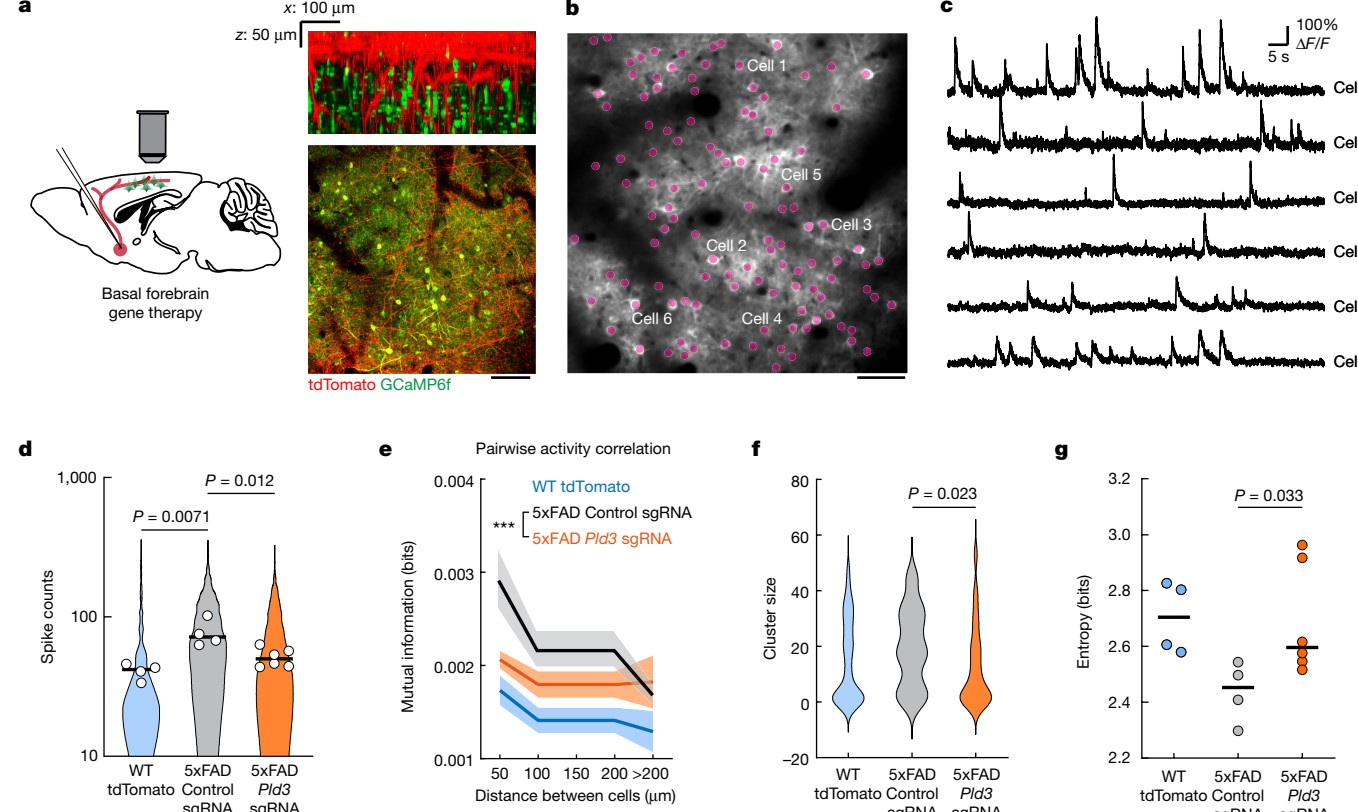

**Fig. 5 | Reduction in axonal spheroids by *Pld3* deletion improves neural circuit function. a**, Schematics of cholinergic neurons in the basal forebrain projecting to the cortex, infected with AAV viruses encoding either *Pld3* or control sgRNAs (left). Right, two-photon images showing intermingled projecting basal forebrain axons (red) from the basal forebrain (red) with GCaMP6f-labelled cortical neurons (green). Scale bar, 100 μm. **b**, Representative two-photon image of GCaMP6f-labelled cortical neurons (purple dots). Scale bar, 50 μm. **c**, Example raw calcium traces from individual cortical neurons. **d**, Single-cell spike counts from individual neurons during a 30 min imaging session. Each dot represents the average spike count from all cells in the same mouse. The violin plots show distributions of spike counts from all individual neurons. **e**, Pairwise mutual information grouped by the

distances between neurons. Data are mean ± s.e.m. Two-way analysis of variance was used to compare between groups. **f**, Neuron cluster size distributions classified by activity patterns (Louvain clustering; Methods). **g**, Quantification of population entropy (a measurement of temporal variance of the firing pattern) from each mouse imaged. For **d**,**e** and **g**, *n* = 4, *n* = 4 and *n* = 6 mice in the WT group, 5xFAD with control sgRNA group and 5xFAD with *Pld3* sgRNA group, respectively. For **f**, *n* = 67, *n* = 21 and *n* = 45 clusters in the WT group, 5xFAD with control sgRNA group and 5xFAD with *Pld3* sgRNA group, respectively. For **d**,**f** and **g**, statistical analysis was performed using one-way analysis of variance to compare among groups and the *P* values indicate a post hoc comparison between the groups, with Sidak's correction for multiple comparisons. For **d** and **g**, the bars indicate the group mean.

their size. The finding that larger PAASs caused more severe conduction blocks was consistent with computational modelling (Supplementary Discussion 2) showing that PAASs resemble electrical capacitors that function as current sinks, and that PAAS size is a major determinant of the degree of conduction defects. Together, our data suggest that the large number of amyloid deposits present in the AD brain have the potential to substantially affect neural networks by widespread disruption of axonal connectivity.

Using in vivo Ca²⁺ imaging, we found that neurons in the cortex of 5xFAD mice showed a pattern of aberrant activity, in agreement with previous reports[36]. We found that these abnormal activity patterns can be rectified by deleting *Pld3* from basal forebrain neurons, which provide cholinergic inputs to cortical neurons. These results raise the possibility that the aberrant neuronal cortical activity could be partly due to conduction defects in long-range cholinergic projections caused by PAASs. The potential relevance of these findings is further highlighted by the observation of neuronal hyperactivity in humans in early stages of AD[38] and the susceptibility of cholinergic neurons to AD neuropathology[35].

In addition to cholinergic projections, strategically located amyloid plaques could have deleterious effects in other brain regions, such as the hippocampus, in which parallel compact axonal bundles follow a

stereotyped projection path along a trisynaptic loop. Memory formation may be particularly vulnerable to axonal conduction delays and blockades, given that memory consolidation depends on the precise timing of hippocampal replay and sharp-wave ripples[39] (Supplementary Discussion 4). Consistent with this view, our quantitative histopathology analysis in a limited number of human post-mortem brain samples from AD or MCI showed that PAAS size and number correlate well with the degree of premortem cognitive decline.

Mechanistically, we found that ELPVs—which probably include MVBs, endolysosomes and autolysosomes—accumulate within axonal spheroids and that their presence is correlated with spheroid size. Moreover, we found an increased presence of ELPVs within spheroids in older 5xFAD mice and in more severely impaired human patients with AD, indicating that ELPV accumulation may be a key feature of disease progression. MVBs are crucial intermediate organelles that evolve through the maturation of endosomes and fuse with autophagosomes and lysosomes[26,31]. Thus, dysregulation in MVB biogenesis has the potential to affect the normal generation of fusion vesicles.

The endosomal sorting complex required for transport (ESCRT) machinery has a major role in MVB biogenesis by regulating the formation of ILVs within MVBs and the sorting of ubiquitinated proteins into ILVs destined for degradation[25]. In contrast to lysosomal-resident

proteins sorted to the limiting membrane of MVBs[32], PLD3 is unique because it is sorted into ILVs through the ESCRT pathway in mammals[3]. Indeed, we found that PLD3 was present within ELPVs and was highly enriched in PAASs, suggesting a potential role of PLD3 in MVB biogenesis. Consistent with this, we found that overexpression of PLD3 in neurons led to a marked enlargement and accumulation of ELPVs and resulted in an overall increase in PAAS size. Furthermore, the human PLD3(V232M) variant was associated with an increased abundance of ELPVs within PAASs, similar to PLD3 overexpression in mice, suggesting that this variant may exert a gain-of-function effect. A study in young *Pld3*-knockout mice showed that classical lysosomes in neuronal cell bodies were enlarged[2], whereas a separate study in HeLa cells showed no lysosomal changes with *PLD3* deletion[3]. Together with our data, this suggests that PLD3 has complex effects on endolysosomes that are age- and context-dependent and that axons are particularly susceptible to endolysosomal abnormalities, especially in the presence of amyloidosis.

Dysfunction of ESCRT components can lead to the enlargement of endolysosomal compartments[3,40]. Given the accumulation of PLD3 in axonal spheroids, it is possible that PLD3 accumulation leads to endolysosomal enlargement by interfering with ESCRT machinery. Consistent with this, we found that PLD3 overexpression led to the enlargement of LAMP1-positive vesicles and the formation of small axonal swellings even in WT mice. However, the substantial enlargement of ELPVs after PLD3 overexpression predominantly occurred in spheroids in the vicinity of amyloid plaques, suggesting that this process is amyloid-dependent. We observed robust endocytic activity at axonal spheroids that was associated with the uptake of Aβ42 into endolysosomal compartments. Moreover, we found that Aβ42 was present within ELPVs at axonal spheroids, consistent with previous immunogold electron microscopy in patients with AD, showing that the most prominent subcellular localization of Aβ42 is within MVBs[41]. Consistent with this, administration of Aβ42 to cultured neurons has been shown to result in MVB enlargement, possibly through interference with ESCRT proteins[33]. Thus, internalization of Aβ from extracellular deposits may be critical for PLD3-induced ELPV accumulation.

Aβ42 accumulation within MVBs may also impair the ubiquitin–proteasome system, leading to defects in protein sorting and ILV invagination[42]. Protein-sorting abnormalities could in turn exacerbate the accumulation of PLD3. Thus, PLD3 could work synergistically with Aβ42 in the same subcellular compartment, leading to greater endolysosomal abnormalities. On the other hand, given that APP and β-site APP-cleaving enzyme (BACE1) also accumulate within PAASs[43], we do not exclude the possibility that intracellularly produced Aβ could also contribute to abnormalities in MVB biogenesis. APP is also sorted into ILVs of MVBs through the ESCRT machinery[40], and deletion of ESCRT components promotes APP processing and increased intracellular Aβ accumulation[40]. Thus, PLD3 and Aβ may constitute a cycle in which axonal endocytosis and/or intracellularly produced Aβ facilitate the generation and accumulation of ELPVs (Extended Data Fig. 12). However, the precise molecular mechanisms explaining the potential interactions between PLD3 and Aβ remain to be elucidated.

Although the focus of this investigation was on PLD3, it is probable that manipulation of other proteins in the endolysosomal pathway could lead to changes in PAAS size. However, given the negligible expression of PLD3 in non-neuronal cells[29] (Extended Data Fig. 7), this molecule could be a promising therapeutic target because global modulation of the endolysosomal pathway may negatively affect glial cells and their roles in controlling protein aggregation and amyloid brain accumulation[44]. Thus, modulation of neuronal MVB biogenesis through PLD3 or other endolysosomal molecules in neurons could constitute strategies for ameliorating PAAS pathology, independent of amyloid plaque removal. Furthermore, although cytoskeletal abnormalities in axons were not the focus of this study, they probably have important roles in the accumulation of vesicles and spheroid enlargement.

Moreover, the presence of hyperphosphorylated tau protein within PAASs as a potential source of tau propagation throughout the neuronal soma[45] suggests important links between plaques, axonal spheroids and neurofibrillary tangles that need further investigation. Finally, in addition to neuronal intrinsic mechanisms, the glial microenvironment around plaques has been shown to have a key role in preventing PAAS formation[46,47]. Thus, the interplay between intrinsic neuronal and extrinsic glial mechanisms may contribute to the formation and enlargement of PAASs and should be considered when designing therapies.

Our findings reveal a cell-intrinsic neuronal mechanism that modulates the size of axonal spheroids and the consequent axonal conduction defects, with important implications for neuronal network dysfunction. Given that axonal spheroids are a prominent feature in various neurological disorders in addition to AD[48–50], it remains to be studied whether spheroids in these conditions share mechanistic properties with those in AD. Thus, our study opens a theoretical and experimental framework for systematically investigating axonal spheroid pathology in a variety of neurological conditions.

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

## Methods

### Mice

5xFAD (34840-JAX, The Jackson Laboratory) mice were used in this study. Rosa26-LSL-Cas9 (026175, The Jackson Laboratory) mice were crossed with 5xFAD mice for CRSIPR–Cas9-mediated gene deletion. The genotyping of 5xFAD mice was performed according to the instructions provided by The Jackson Laboratory. All of the animal procedures were approved by the Institutional Animal Care and Use Committee at Yale University.

### Antibodies and reagents

The following primary antibodies were used in this study: anti-LAMP1 (DSHB, 1D4B, 1:200), anti-GFP (Aves Labs, GFP-1020, 1:500), anti-cathepsin D (Abcam, EPR3057Y, ab75852, 1:250), anti-ATP6V0A1 (Thermo Fisher Scientific, PA5-54570, 1:200), anti-APP (Thermo Fisher Scientific, LN27, 13-0200, 1:100), anti-PLD3 (Sigma-Aldrich, HPA012800, 1:200), anti-Aβ(1–42) (Abcam, ab10148, 1:200), anti-Aβ(1–42) (Abcam, mOC98, ab201061, 1:200), anti-MAP2 (Abcam, ab5392, 1:200), anti-IBA1 (Novus Biologicals, NB100–1028, 1:200), anti-S100B (R&D Systems, AF1820, 1:100), anti-ALDH1L1 (NeuroMab, P28037, 1:250) and FM 1–43 (Life Technologies, F35355). All secondary antibodies used were conjugated with Alexa dyes from Thermo Fisher Scientific (1:500). Thioflavin S (Sigma Aldrich, T1892) was used to stain amyloid plaques in fixed tissue. FSB (Santa Cruz, CAS 760988-03-2) was used to label plaques in live mice. To study spheroid endocytosis of extracellular Aβ, fluorescently labelled Aβ(1–42) (AnaSpec) was used. Pitstop2 (Abcam, ab120687) was used to inhibit endocytosis.

### AAV production and delivery

The GCaMP6f and GCaMP6s viruses were purchased (UPenn Virus Core, AV-9-PV2822 and AV-9-PV2824; Addgene, 100837 and 100843). The ASAP3 construct was purchased from Addgene (Addgene, 132331). Customized AAV vectors for overexpression were constructed based on plasmid 28014 from Addgene, in which the *GFP* sequence was deleted and replaced by the customized sequences described below. In many cases in which the virus transduced both a target protein and a fluorescent protein reporter, a *GFP* without the stop codon and P2A sequence was placed in front of the target protein sequence in the same open reading frame, as described previously[51]. The target proteins used in this study were as follows:

*tdTomato*: sequence is available online (http://www.tsienlab.ucsd.edu/Samples/PDF/tdTomato-map%20&%20sequence.pdf), synthesized by Integrated DNA Technologies; *mCherry-SEpHluorin*: the sequence was cut from Addgene 32001; *Lamp1-GFP*: the *Lamp1* sequence was amplified from mouse brain mRNA, using the 5′ primer TGCGTCGCGCCATG-GCGGCC and 3′ primer GATGGTCTGATAGCCGGCGT; *GFP-P2A-Pld3*: the *Pld3* sequence was amplified from mouse *Pld3* cDNA (GE open biosystem), using the 5′ primer ATGAAGCCCAAACTGATGTACCAGG and the 3′ primer TCAAAGCAGGCGGCAGGC. The sgRNA constructs for *Pld3* deletion were cloned using plasmid 60229 from Addgene. The sequences of the sgRNAs were: *Pld3* sgRNA 1: GTCCTGATCCTGG-CGGTAGT; *Pld3* sgRNA 2: GCTAGTGGAGGGGTTGCTCG and control scrambled sgRNA: GGAAGAGCGAGCTCTTCT. All of the constructs were verified by DNA sequencing, and the expression or deletion of the target proteins was tested using immunohistochemistry.

AAV2 viruses were produced in HEK293T cells (American Type Culture Collection (ATCC)) and purified according to previously described procedures[52] using a two-plasmid helper-free system (PlasmidFactory). The virus titre was determined by counting the infection in HEK293 cells. AAV vectors were injected into the subarachnoid space in one hemisphere as previously described[51]. The total viral particles injected per mouse was approximately $10^7$.

### Cranial window implant

Eight-month-old 5xFAD mice were anaesthetized with ketamine/xylazine solution (100 mg per kg and 10 mg per kg, respectively) and hair was removed from the skull area. Buprenex (0.1 mg per kg), dexamethasone (2 mg per kg) and carprofen (5 mg per kg) were given subcutaneously at this point. The mouse was placed on a heating pad during the surgery and anaesthesia was checked periodically. Povidone-iodine solution was applied to the skin and cleaned with ethanol, and eye ointment was applied to the eyes. A small piece of skin was removed to expose the skull, and the membranous layer on the skull surface was removed by forceps. A 4 mm diameter circle was drilled on the contralateral hemisphere of virus infusion (approximate location of the centre is −2.5 mm from bregma and 2.5 mm from the midline). The skull was rinsed with sterile PBS periodically to avoid excessive heating. The skull was thinned in a circumferential area and then lifted with fine forceps without causing injury to the underlying pila surface. Gelfoam sponge (Pfizer) was used to absorb blood after lifting the skull. Using a pair of very fine forceps, the dura was removed within the circle area and a 4 mm cover glass was gently pressed onto the brain surface and glued to the skull. A customized head bar was glued (for acute imaging) or chronically implanted (with dental cement, for chronic imaging) onto the skull. For chronic imaging, mice were placed onto a heating pad to recover after the surgery and given buprenex (0.1 mg per kg) and carprofen (5 mg per kg) for 3 days. Imaging procedures started 1 month after the surgery.

### In vivo two-photon imaging

In vivo imaging was performed using a two-photon microscope equipped with a Ti-sapphire tuneable laser (Spectra Physics), a gallium arsenide phosphide (GaAsP) detector (Prairie technology) and a ×20/1.0 NA water-immersion objective (Leica), or the Ultima Investigator multiphoton microscope (Bruker) with the Insight X3 tuneable ultrafast laser (Spectra Physics) and a ×20/1.0 NA water-immersion objective (Olympus), using the associated Prairie View software. GFP was excited at 920 nm; dTomato and tdTomato were excited at 920 nm/1,045 nm; and FSB was excited at 850 nm. For chronic imaging, a location close to the centre of the cranial window was selected as the starting point and the blood vessel pattern was recorded. The coordinates of each ROI were recorded as well. To relocate in the next imaging session, the starting point was relocated on the basis of the recorded coordinates and the field of view was adjusted to match the recorded blood vessel pattern.

### Aβ(1–42) preparation and injection

Fluorescently tagged Aβ(1–42) peptide (AnaSpec, 60480-1) was reconstituted in DMSO to a final concentration of 1 mg ml$^{-1}$. The solution was 1:10 (v/v) diluted in fresh artificial cerebrospinal fluid before being injected into the subarachnoid space as previously described[53]. We injected 10 μl of Aβ(1–42) solution into each mouse and collected the brain the next day. Brain tissue was prepared for immunohistochemistry analysis.

### Calcium imaging of cortical axons and related analysis

Six-to-eight-month-old 5xFAD mice were injected with GCaMP6 virus through the subarachnoid space on one hemisphere to label cortical neurons and measure local axonal conduction properties. For measurements of interhemispheric axonal conduction, GCaMP6f virus was injected stereotaxically with the following coordinates: bregma (AP, −0.34; ML, 1.65; DV, 0.45; angle, 0°)[54] (Allen Mouse Brain Connectivity Atlas (2011)). After more than 2 weeks of the injection, an acute cranial imaging window was implanted onto the contralateral hemisphere as described above. For stimulated calcium imaging, an additional opening on the skull was made on the ipsilateral side of the virus infusion. A glass electrode was inserted through this opening using a motorized micromanipulator and used for electrical stimulation.

The ROI was identified under a two-photon microscope. GCaMP6-labelled neurons were imaged through excitation at a wavelength of 920 nm. A limited field of view was used to improve

the sampling rate. GCaMP6s was imaged at 2 Hz and GCaMP6f was imaged at 10 to 20 Hz. Only axons that displayed spontaneous calcium transients at least once per minute were used for analysis. For stimulated calcium events, mice were anaesthetized using 0.5% isoflurane. Stimulation trains of 2 ms pulses were delivered to the glass electrode at 50 Hz (18 ms interval) with 10 to 60 µA currents for 500 ms. The calcium responses within the imaging window were monitored after stimulation. The stimulating electrode was adjusted to different depths within the cortex to trigger responses in contralateral hemisphere axons. Three consecutive trials were acquired for each axon and the responses were averaged.

The raw GCaMP6 fluorescence intensity was normalized to $\Delta F/F$ for analysis. Images were then spatially smoothed with a $3 \times 3$ window. Several ROIs were selected on each axon. The average $\Delta F$ before stimulation was used as the baseline measurement. To estimate the calcium rise time (a surrogate for AP spike time[55]), the calcium trace from the event-specific peak to the first data point exceeding the baseline was used as the rising phase of the event. This rising phase trace was fitted to an exponential equation: $Y = 1 - \exp(-k \times (x - t))$. The spike timing estimation ($t_0$) was then calculated by extrapolating the $x$-intercept. For analysis of the spontaneous $Ca^{2+}$ transients, we calculated the correlation coefficients between two ROIs chosen on each axonal side of a particular spheroid using the Pearson correlation coefficient.

## Calcium imaging of cortical neuronal networks and related analysis

AAV2 viruses encoding either *Pld3* sgRNA or control sgRNA were injected stereotaxically into the basal forebrain of 5xFAD mice (aged 6 to 8 months) with the following coordinates: bregma (AP, 0.62; ML, 1.2; DV, 4.85; angle, 0°)[54] (Allen Mouse Brain Connectivity Atlas (2011)). GCaMP6 virus was injected through the subarachnoid space on the ipsilateral hemisphere of basal forebrain to label cortical neurons.

The ROI with intermingled projecting axons from the basal forebrain and GCaMP6f-labelled cortical neurons was identified under a two-photon microscope. Calcium imaging was performed in cortical neurons of awake mice in the same region as those projecting forebrain axons. Spontaneous calcium activities of neurons were recorded at 25 Hz for 30 min.

To analyse the cortical neuron activities, we applied a rigid motion correction[56] to the raw time-lapse data and segmented the neuronal signal using a previously established algorithm based on robust estimation[57]. Cell locations were defined as the centroid points from the spatial masks of each cell. We further estimated the neuronal spikes on the basis of calcium traces as previously shown[58]. To calculate the pairwise mutual information, we used previously described methodologies[59]. Mutual information from each cell pair in the same imaging session was grouped by the distances between neurons. To calculate neuron clusters with similar activity patterns, we used previously described graph-theory-based community detection algorithms[60], with forced deterministic behaviour for better reproducibility, and determined the sizes of each cluster of cells identified in a single imaging session. Calculation of population entropy as performed according to a previously reported method[61]. To account for different numbers of cells imaged, we drew a random subsample of 100 neurons from each mouse and calculated the population entropy. This process was repeated 100 times and the average results were recorded.

## Voltage imaging and analysis

AAV2-Syn-ASAP3 virus was injected stereotaxically into 5xFAD mice (aged 6 to 8 months) with the following coordinates: bregma (AP, −0.34; ML, 1.65; DV, 0.45; angle, 0°)[54] (Allen Mouse Brain Connectivity Atlas (2011)). After more than 2 weeks of the injection, an acute cranial imaging window was implanted onto the ipsilateral hemisphere as described above. An additional opening on the skull was made on the contralateral side of the virus infusion. A glass electrode was inserted through this opening using a motorized micromanipulator and used for electrical stimulation.

The ROI was identified under a two-photon microscope. Line scan imaging of ASAP3-labelled neuronal soma was performed through excitation at a wavelength of 920 nm. The scanning speed was set to 1 kHz. Stimulation trains of 5 ms pulses were delivered at 10 Hz (95 ms intervals) with 10 to 100 µA currents for 1 s.

Line scan images were used to extract the voltage response of ASAP3. On each line of the kymograph, the intensity of the pixels covering the ASAP3-labelled neuronal soma was averaged to represent the membrane potential of the neuron at given time, which was used to generate the voltage response trace later. The response trace showed a large dip after each electrical stimulation pulse. The time of the dip indicated the antidromic AP time[62] on the neuronal soma. To quantify the AP conduction failure, we performed FFT of the voltage response curve and took the 10 Hz (the same frequency as the electrical stimulation) component power as an indicator. The amplitude of the electrical current pulse was gradually increased; and the FFT power–current amplitude could be plotted. We fit the plot with a logistic function: $Y = a/(1 + \exp(-k \times (x - b)))$. The current amplitude at the half-height of the logistic curve was defined as the threshold of the electrical stimulation.

## Human post-mortem brain tissues

Formalin-fixed human post-mortem brain tissue blocks were acquired from brain banks, with ethics approval from each providing organization. The middle frontal gyrus, a cortical region affected in early stages of Alzheimer's disease[63], was used for this study. Detailed information is provided in Extended Data Fig. 4, including 12 cases of AD, four cases of AD with *PLD3* variants and six cases of MCI. Cases were matched for age, sex and *APOE* genotype for most quantifications except for PLD3(V232M)-variant samples. For immunohistochemistry analysis of human tissue, 30-µm-thick slices were prepared and treated with sodium citrate solution at 95 °C for 45 min, before staining with primary antibodies for 3 days.

## Acute organotypic brain-slice culture

Brain-slice cultures were prepared from 5xFAD mice (aged 8 months) according to a previously established protocol[64]. In brief, we dissected the hippocampal region in a sterile hood from anaesthetized mice. We then manually cut coronal sections approximately 300 µm thick and transferred the slices onto a Millicell culture membrane (Thermo Fisher Scientific, PICM03050). The culture membrane was placed in a six-well plate filled with 1 ml of the culture medium, and was placed in an incubator at 37 °C under 5% $CO_2$. We examined the slice condition after 7 days with a light microscope. Healthy slices were then used for experiments. To measure endocytosis, we added endocytosis marker FM 1-43 dye (Thermo Fisher Scientific, T35356) to the culture medium to a final concentration of 1 µM. We incubated the slices with the dye for 1.5 h and then washed the slices with fresh medium twice before fixing the slice in 4% paraformaldehyde. For experiments blocking endocytosis, we dissolved Dynasore (Tocris Bioscience) or the endocytosis inhibitor PitStop2 (Abcam, ab120687) in dimethyl sulfoxide (DMSO) and added it to the culture medium with different final concentrations of FM 1-43 dye. The control groups used DMSO with no drug following the same dilutions.

## Plaque-associated axonal spheroid imaging and quantification

Fixed-tissue imaging was performed using a confocal microscope (Leica SP5 or Lecia SP8), and the images were taken using a ×63/1.4 NA oil-immersion objective (Leica). Individual PAAS sizes were measured on the basis of LAMP1, APP or V0A1 immunohistochemistry. Specifically, we examined the cross-sections of the target PAASs through the high-resolution $z$-stack and selected the outlines of individual cross-sections using NIH/Fiji software. The largest cross-sectional area was recorded as the PAAS size. For quantification of the proportion

of PAASs with ELPVs, we defined PAASs in a binary manner as either having or not having at least one obvious LAMP1-positive enlarged ring, consisting of a clear LAMP1-labelled circular outline and a darker lumen with an area greater than 0.25 μm², as measured using NIH/Fiji. To classify PAASs into neutral versus acidic, using the genetically encoded pH sensor SEpHluorin[65], we established an arbitrary threshold of green/red fluorescence intensity ratio of 0.75. To classify PAASs into cathepsin D low versus high groups using immunofluorescence, we used an arbitrary threshold of 100 fluorescence intensity as measured using NIH ImageJ/Fiji.

## Expansion microscopy

Expansion of brain sections was performed according to conventional immunostaining. Brain sections were treated with glutaraldehyde (TCI Chemicals, G0068) and then processed for gelation, digestion and expansion, as described previously[66,67]. In brief, brain sections were first incubated with monomer solution (1× PBS, 2 M NaCl, 8.625% (w/w) sodium acrylate, 2.5% (w/w) acrylamide, 0.15% (w/w) N,N′-methylenebisacrylamide) at 4 °C for 45 min. The sections were then transferred into a gel chamber and incubated in gelling solution (concentrated stocks (10%, w/w) of ammonium persulfate initiator and tetramethyl-ethylenediamine accelerator added to the monomer solution for up to 0.2% (w/w) each and the inhibitor 4-hydroxy-2,2,6, 6-tetramethylpiperidin-1-oxyl added up to 0.01% (w/w) from a 0.5% (w/w) stock) at 37 °C for 1.5–2 h for gelation. The gels were then fully immersed in proteinase solution (proteinase K (New England Biolabs, P8107S) diluted 1:100 to 8 U ml⁻¹ in digestion buffer (50 mM Tris (pH 8), 1 mM EDTA, 0.5% Triton X-100, 1 M NaCl)) at 37 °C overnight. Digested gels were next placed in excess volumes of double deionized water (double-distilled H₂O) for 25 min to expand. This step was repeated 3–5 times in double-distilled H₂O, until the size of the expanding sample plateaued.

## Transmission electron microscopy

Twelve-month-old 5xFAD mice were perfused with 4% PFA and the brain tissues were sectioned into 50-μm-thick slices using a vibratome (VT1000S, Leica). The slices were refixed in 2% glutaraldehyde in 0.1 M cacodylate buffer (pH 7.4) for 1 h, then post-fixed in 1% OsO₄ in the same buffer at room temperature for 1 h. After en bloc staining with 2% aqueous uranyl acetate for 30 min, tissues were dehydrated in a graded series of ethanol to 100%, followed by propylene oxide and finally embedded in EMBed 812 resin. Tissue blocks were polymerized overnight in an oven at 60 °C. Thin sections (60 nm) were cut by a Leica ultramicrotome (UC7) and post-stained with 2% uranyl acetate and lead citrate. Sample grids were examined on the FEI Tecnai transmission electron microscope with an accelerating voltage of 80 kV, and digital electron micrographs were recorded with an Olympus Morada CCD camera and iTEM imaging software.

## AAV-mediated molecular manipulations

AAV vectors were injected into approximately 3-month-old and approximately 7-month-old 5xFAD mice. AAVs were infused through the subarachnoid space as previously shown[51]. Brain tissues were collected around 1.5 months and 3 months after virus injection for treatments initiated at 3 and 7 months of age, respectively, and fixed with 4% paraformaldehyde. Brain slices of 50 μm thickness were prepared and stained with anti-LAMP1 antibodies (DSHB, 1D4B) and thioflavin S.

Imaging and quantification of PAASs were performed as described previously[51,66]. In brief, tiled z-stack images of the infected cortical regions were taken at zoom 1 and 1 μm z-steps. Individual plaques were segmented from the tiled images and blinded to the mouse and treatment information. For measurement of PAAS bulb size, the z plane with the largest cross-sectional area of each individual spheroid was selected, and the cross-sectional area was measured using NIH ImageJ/

Fiji by manually selecting the outlines of that cross-section based on the outline of virally expressed cytoplasmic GFP fluorescence.

## Computational modelling

All modelling experiments of axon spheroids used methodologies that were previously validated[68,69]. The chosen morphology parameters and ion-channel distributions were held constant within each compartment for each simulation. The axon length was 566 μm and diameters were set from 0.1 to 0.9 μm. The spheroids were modelled as a 'cylinder and stick'. The stick (5 μm length and diameter varying from 0.3 to 8.3 μm) was connected to the middle of the axon and on the other end to the cylinder (a bulb head) of which the surface area varied from 0 to 7,100 μm² (equivalent spherical diameter varied from 0 to 150 μm). In most simulations, the axon contained voltage-gated sodium and potassium channels and a leak current. The densities of the voltage-gated channels were varied from 0 to an amount 1.1× as strong as needed to produce regenerative (sustained) APs in the axon. Channel conductances were set in the spheroids with the same (varying) strength as was present in the axon. One or more strong-current injections (0.2 ms, 2 nA) were applied to one end of the axon to reliably evoke a single or multiple input AP(s) that subsequently propagated to the spheroids. Current injections just over the threshold generated qualitatively similar results (larger diameter axons required more current to evoke an AP). We checked for the presence of output APs on the other side of the spheroids by testing for voltages exceeding 10 mV. We counted the number of output APs that were present in a 10 s simulation where either the single input AP occurred at, or the train of 20 Hz input APs began at 100 ms.

## Statistics and reproducibility

The methods to analyse ELPVs, cathepsin D levels or the pH in individual axon spheroids, as well as spheroid size after treatment, the specific numbers of plaques or spheroids measured from an individual human or mouse sample for each experiment are provided in the figure legend of the particular experiment, and the average results were used as the representative outcome for that individual. The number of post-mortem human tissues or mice was used as the sample size in these cases. To analyse differences in the calcium rise time between ROIs along axon segments, the number of axons was used as the sample size. This is because individual in vivo experiments had very few axons measured due to the necessary sparse labelling method used in these experiments. To analyse the interhemispheric calcium rise time delay, both the number of axons and the number of mice were used as sample sizes. For analysis of interhemispheric conduction using a voltage sensor, the number of cell bodies were used as the sample size. For analysis of neural circuit function, the number of mice was used as the sample size.

In all graphs, individual data points are shown. In all statistical comparisons, non-parametric tests were used unless otherwise justified. Specific tests used for each graph can be found in the corresponding figure legend. When more than two groups were considered and compared, corrections for multiple comparisons were performed as part of the post hoc analysis. All statistical analysis was performed using GraphPad Prism.

Representative confocal images shown in the figures were repeated independently by at least two experimenters in multiple mice or human samples.

## Inclusion and ethics statement

We balanced both sexes in all of our experiments to the best of our ability. No sex-specific effects of axonal spheroids on conduction, or modulation of PLD3 on the degree of axonal spheroids were observed in our study. The brain tissues were obtained from the Sun Health Research Institute Brain and Body Donation Program, the Mayo Clinic, the University of Washington Alzheimer's Disease Research Center

Neuropathology Core and the Alzheimer's Disease Research Center at Washington University in St Louis, in complete compliance with the human tissue research use regulation in each organization.

## Reporting summary

Further information on research design is available in the Nature Portfolio Reporting Summary linked to this article.

## Data availability

The raw datasets generated and analysed during this study are not deposited to a publicly accessible repository due to the large amount of storage required but are available from the corresponding author on reasonable request.

## Code availability

All of the custom codes for FIJI and MATLAB used in this study have been deposited at GitHub (https://github.com/PaulYJ/Axon-spheroid). The scripts are free to the public to view, distribute and use. Our NEURON simulation environment computer code is available at ModelDB (http://modeldb.yale.edu/187612).

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

**Acknowledgements** This project was supported by National Institute of Health grants RF1AG058257, R01NS115544 and R01NS111961 (to J.G.); and a Cure Alzheimer's Fund Research Grant (to J.G.). P.Y. was funded by Shanghai Municipal Science and Technology Major Project, the Lingang Laboratory (grant no. LG-QS-202203-09) and Shanghai Natural Science Foundation (22ZR1415000). Y.C. was supported by a Brightfocus Foundation fellowship. Computational modelling was supported by National Institute of Health grant R01DC009977 (to T.M.M.) and T15LM007056 (to R.A.M.). We are grateful for the use of the Louise computer cluster at the Yale Center for Research Computing and Center for High Performance Computation in Biology and Biomedicine at Yale University (supported by NIH grants RR19895 and RR029676-01) and the Neuroscience Gateway (NSG) Portal supported by the National Science Foundation[70]. D.C. was funded by National Institute of Health grant R25NS079193 and the NIH Loan Repayment Program Award. We thank the staff at the Sun Health Research Institute Brain and Body Donation Program of Sun City, Arizona for the provision of human brain tissues, supported by the NINDS (U24 NS072026), the NIA (P30 AG19610), the Arizona Department of Health Services (contract 211002), and the Arizona Biomedical Research Commission (contracts 4001, 0011, 05-901 and 1001). Additional MCI and AD brain tissue was provided by D. W. Dickson and M. Deture, and D. Keene and T. Bird (supported by NIA P50AG05136). We thank the staff at the Alzheimer's Disease Research Center at Washington University at St Louis for providing the human AD tissue with the *PLD3* variant, funded by the following grants: Healthy Aging and Senile Dementia (P01 AG003991), Alzheimer's Disease Research Center (P30 AG066444) and Adult Children Study (P01 AG026276); the staff at the Center for Cellular and Molecular Imaging, Electron Microscopy Facility at Yale Medical School for assistance with the electron microscopy experiments; and P. Coish for reading and editing the manuscript.

**Author contributions** P.Y. and J.G. conceived the study. P.Y., M.Z., L.T. and J.G. designed the study. P.Y. performed time-lapse in vivo imaging and spontaneous calcium imaging. L.T. and M.Z. designed and performed stimulated calcium imaging, interhemispheric calcium imaging, calcium imaging of neural networks and voltage imaging. M.Z., L.T. and H.D. performed calcium imaging after molecular manipulation. T.M.M., R.A.M. and P.Y. designed and performed computational modelling experiments. P.Y. and M.Z. performed confocal microscopy and quantitative histology. M.Z. designed and performed PLD3-related experiments. L.T. and H.D. performed expansion microscopy. Y.C. performed electron microscopy. P.Y. performed the endocytosis experiments. D.C. performed quantitative histology. P.Y., M.Z., L.T., J.G. and T.M.M. analysed data. P.Y., M.Z., L.T. and J.G. prepared the manuscript. J.G. supervised the study.

**Competing interests** J.G. is a member of the scientific advisory board at Vigil Neuro. The other authors declare no competing interests.

**Additional information**
**Correspondence and requests for materials** should be addressed to Jaime Grutzendler.

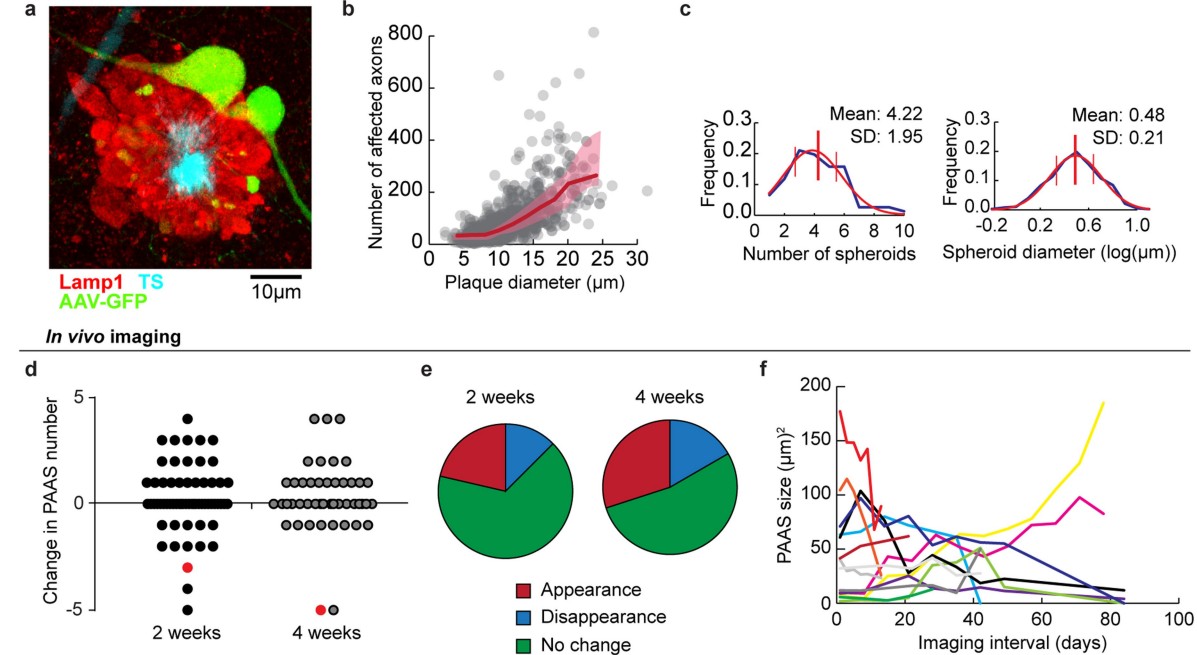

**Extended Data Fig. 1 | Amyloid plaque-associated axonal spheroids predominantly show structural stability and some dynamism over extended intervals. a**, Confocal image of spheroids labelled with anti-LAMP-1 in a 5xFAD mouse. An axon labelled by GFP-expressing AAV co-localizes with LAMP-1. **b**, Estimation of the total number of spheroid-affected axons around individual amyloid plaques (Supplementary discussion 1). **c**, Frequency distribution of the number of spheroids per individual axon (top graph) and logarithmic transformation of the diameters of individual spheroids (bottom graph) quantified from confocal images of virally labelled individual axons, show a gaussian distribution (D'Agostino & Pearson normality test > 0.05,

n = 76 for bulb number and 382 for bulb size. The fitted gaussian curves, 25 and 75 percentile values are marked with red lines). **d**, Quantification of the changes in axon spheroid number at different time intervals from *in vivo* time lapse images of individual axons, labelled with AAV-tdTomato (see Fig. 1). Each dot indicates an axon. Red dots indicate observed spheroid disappearance events. **e**, Pie chart representation of data in panel **d**, showing the proportions of imaged axons that showed PAAS appearance, disappearance, or no change during the respective time intervals. **f**, PAAS sizes change over time in individual axonal segments traced by *in vivo* imaging. Each line indicates a single axon.

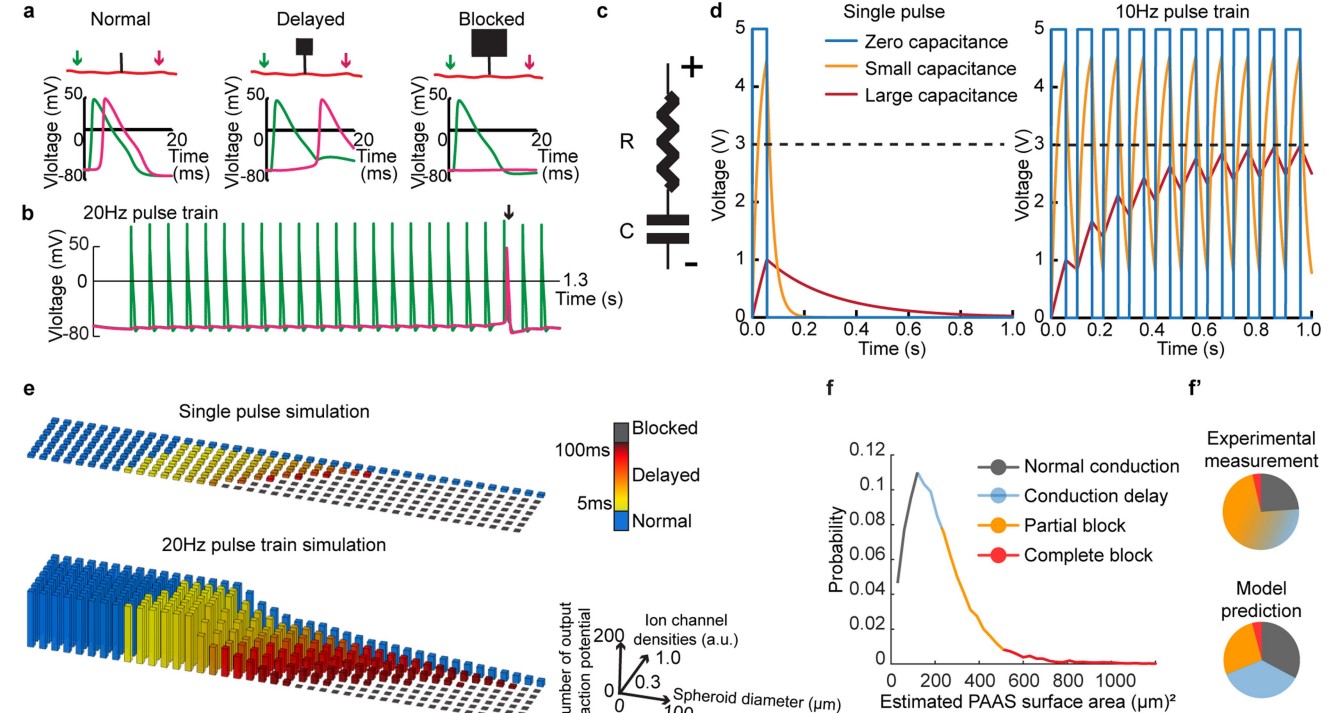

**Extended Data Fig. 2 | Computational modelling of axonal conduction abnormalities caused by PAAS. a**, Computer simulations of membrane potentials recorded at two points on each side of PAAS (green and magenta arrows in upper panels) during a single action potential. Three different scenarios are presented demonstrating PAAS size-dependent conduction delay or block (lower panels) (See Supplementary Discussion 2 for details of the modelling results). **b**, Computer simulation of membrane potentials recorded at two points on each side of PAAS (green and magenta arrows) during a 20 Hz stimulation train. While single action potentials can be completely blocked by larger PAAS, repetitive stimulation can eventually lead to successful

conduction of the action potential due to a capacitor effect of PAAS (Supplementary Discussion 2). **c-d**, Modelling of a simple resistor-capacitor electric circuit with 3 different levels of capacitance. Dashed line indicates 3 volts as an arbitrary threshold mimicking the minimal membrane potential to trigger neuronal firing. **e**, Representation of the simulation results with a range of spheroid diameters and membrane ion channel densities. **f**, Estimated probability distribution of the degree of conduction disruption in PAAS-forming axons. **f'**, Pie charts showing percentages for different types of conduction disruption patterns observed experimentally or by computational model prediction.

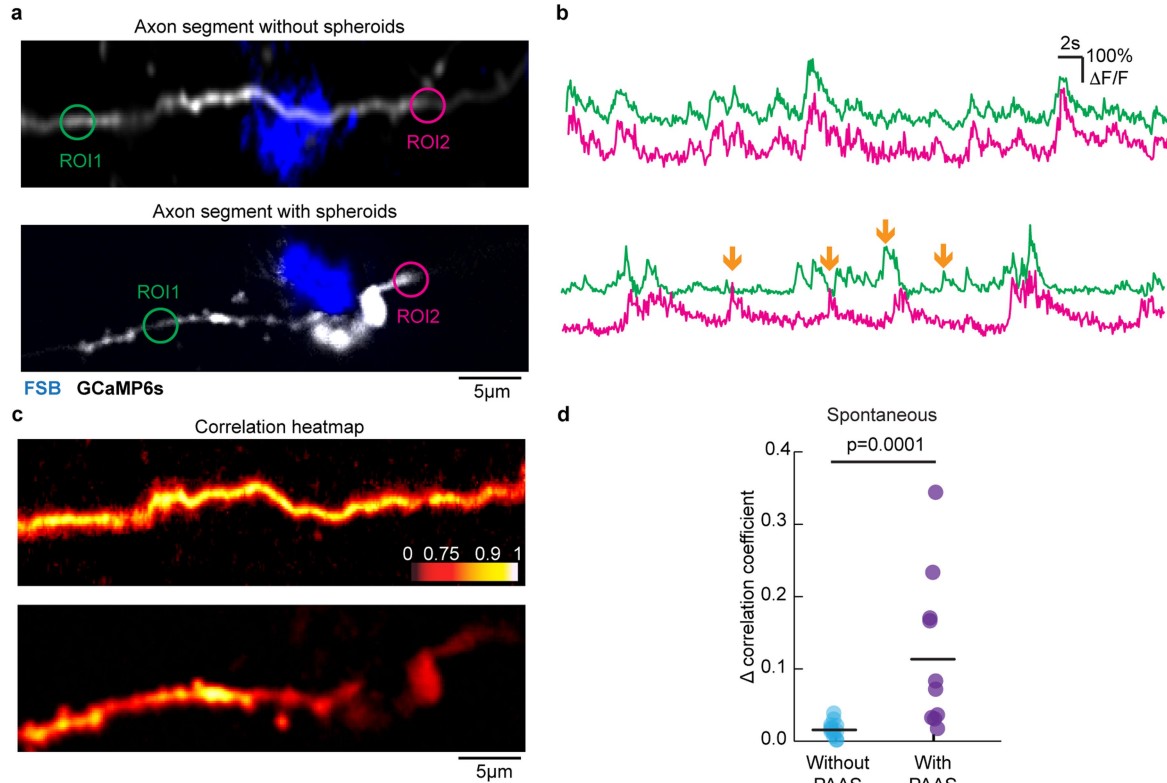

**Extended Data Fig. 3 | Axonal spheroids markedly disrupt spontaneous action potential conduction. a**, Two-photon *in vivo* calcium imaging of spontaneous activity in axons near amyloid plaques (blue) with and without PAAS. Given the relatively low frequency of spontaneously active neurons that can be captured in the vicinity of plaques, we used lower frame rates to image larger fields of view and were thus unable to measure precisely the $Ca^{2+}$ rise times like in Fig. 1. **b**, Example traces of GCaMP6s fluorescence signal were obtained from ROIs (green and magenta circles) at the two sides of the plaques indicated in **a** (2Hz imaging frame rate). Mismatched $Ca^{2+}$ transients are indicated with orange arrows. **c**, Correlation maps were calculated using the average fluorescence intensity within ROI1 (green circle in **a**, as reference, and colour-coded for correlation coefficient to every other pixel within the field of view. **d**, Decorrelation of GCaMP6s fluorescence in ROIs at the two axonal sides with respect to the plaque, during spontaneous $Ca^{2+}$ transients (n = 12 axons without PAAS; and n = 10 axons with PAAS). Two-tailed Mann-Whitney test was used for comparison.

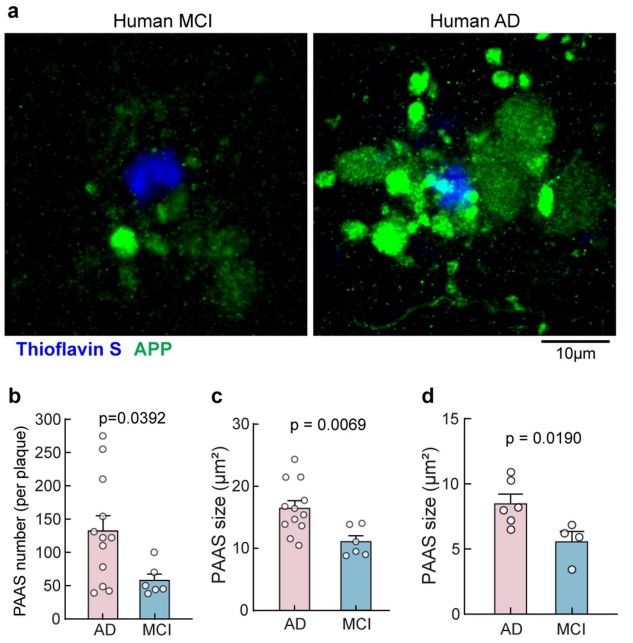

**a**

Human MCI Human AD

**Thioflavin S APP**

10μm

**b**

PAAS number (per plaque)

p=0.0392

*Quantification with APP*

AD MCI

**c**

PAAS size (μm²)

p = 0.0069

*Quantification with APP*

AD MCI

**d**

PAAS size (μm²)

p = 0.0190

*Quantification with V0A1*

AD MCI

**e**

| AD & MCI patients w/o PLD3 variant | | | | |
|---|---|---|---|---|
| Subject | Disease status | ApoE | Age | Braak |
| #1375 | AD | 3/4 | 77 | 5 |
| #1406 | AD | 3/4 | 85 | 4 |
| #1353 | AD | 3/4 | 92 | 5 |
| #1354 | AD | 3/3 | 85 | 4 |
| #C20 | AD | 3/4 | 74 | 6 |
| #1335 | AD | 3/3 | 76 | 5 |
| #C15 | AD | 3/3 | 59 | 5 |
| #C21 | AD | 3/4 | 74 | 5 |
| #0189 | AD | 3/4 | 88 | 5 |
| #0195 | AD | 3/4 | 83 | 5 |
| #1981 | AD | 3/3 | 88 | 3 |
| #C22 | AD | 3/4 | 83 | 5 |
| #0233 | MCI | 3/3 | 85 | 3 |
| #0452 | MCI | 3/4 | 85 | 3 |
| #0512 | MCI | 3/4 | 88 | 3 |
| #9822 | MCI | 3/3 | 90 | 3 |
| #1953 | MCI | 3/4 | 88 | 4 |
| #0550 | MCI | 3/4 | 89 | 5 |

**f**

| PLD3 V232M AD patients | | | | | | |
|---|---|---|---|---|---|---|
| Subject | Disease status | Age | CDR | Amyloid (A) | Tangles (B) | Plaques (C) |
| #60591 | AD | 79 | 3 | 3 | 3 | 3 |
| #60549 | AD | 75 | 3 | 3 | 3 | 3 |
| #60919 | AD | 86 | 3 | 3 | 3 | 3 |
| #62558 | AD | 79 | 3 | 3 | 3 | 3 |

**Extended Data Fig. 4 | PAAS numbers in humans correlate with severity of cognitive impairment. a**, Axon spheroids labelled by Amyloid Precursor Protein (APP) immunofluorescence in post-mortem human brain (middle frontal gyrus), from subjects with mild cognitive impairment (MCI) and AD. **b**, Total spheroid number around individual plaques using APP immunostaining. Each dot indicates the average of 25 plaque measurements of an individual subject. Bars indicate group average. N = 12 AD and 6 MCI subjects. **c**, Spheroid size in AD and MCI patients based on APP immunofluorescence. Each dot represents an average measurement of 50 PAAS. N = 12 AD and 6 MCI subjects. **d**, Spheroid size in AD and MCI patients based on V0A1 immunostaining. Each dot represents an average measurement of 200-250 PAAS. N = 6 AD and 4 MCI patients. **e**, ApoE genotype, age, and Braak stage information of human AD and MCI brain tissue used in this study. **f**, CDR score and ABC score of patients with human AD with PLD3 V232M variant used in the study. Two-tailed Mann-Whitney tests were used in **b**, **c** and **d**. Data are represented as mean ± S.E.M.

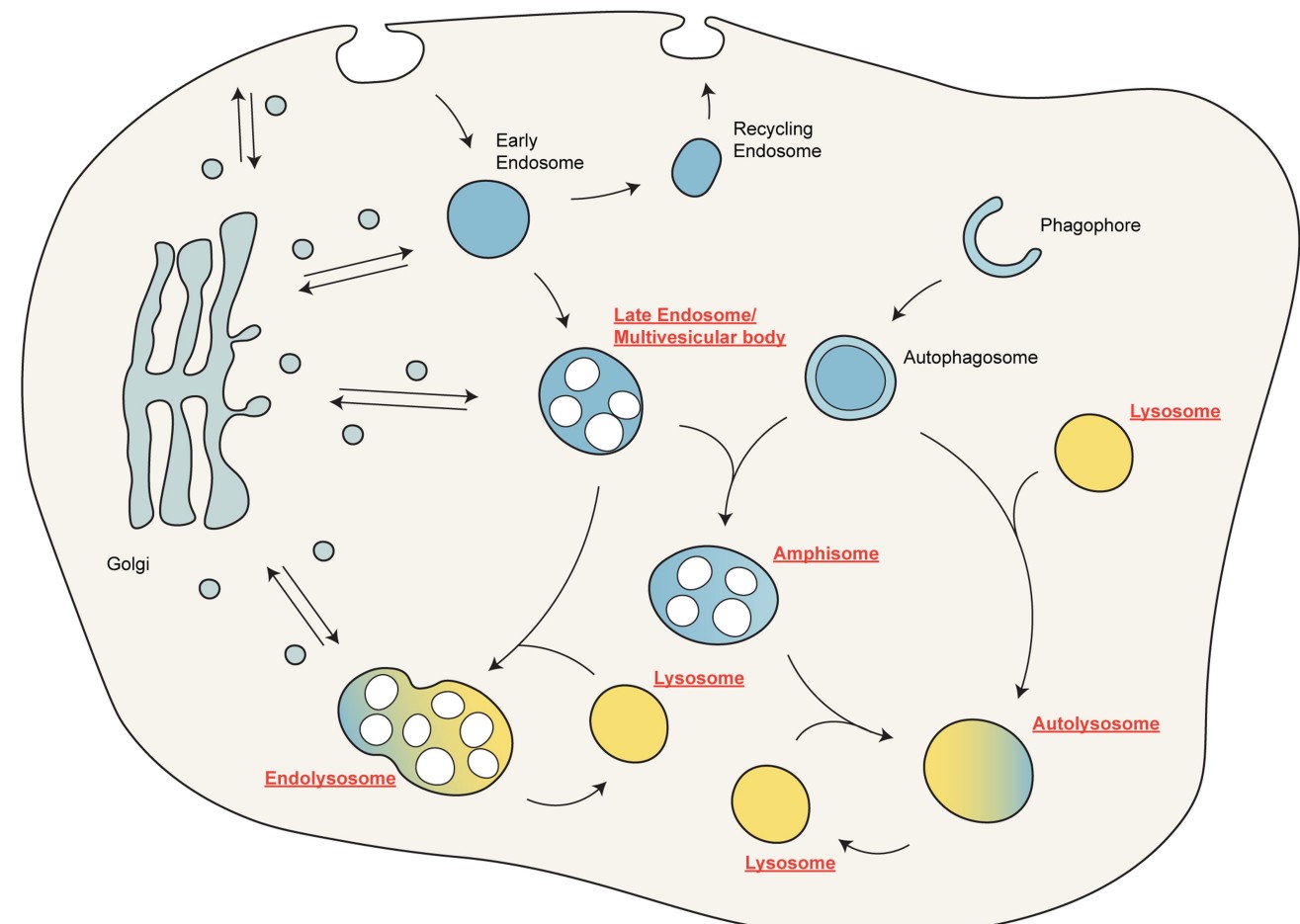

**Extended Data Fig. 5 | Schematic diagram of the potential source of enlarged LAMP1-positive vesicles through disruption of the endosomal-lysosomal-autophagic system.** The endosomal-lysosomal-autophagic system is a dynamic and interconnected network of membranous organelles and vesicles that involves multiple biological processes, including endocytosis, autophagy, organelle maturation, vesicle trafficking and degradation. Enlarged LAMP1-positive vesicles (ELPVs) likely represent vesicular organelles at different stages of maturation into lysosomes, including late endosomes/multivesicular bodies (MVBs), endolysosomes (fusion product between late endosomes/MVBs and lysosomes), amphisomes (fusion product between autophagosomes and late endosomes/MVBs) and autolysosomes (fusion product between autophagosomes and lysosomes).

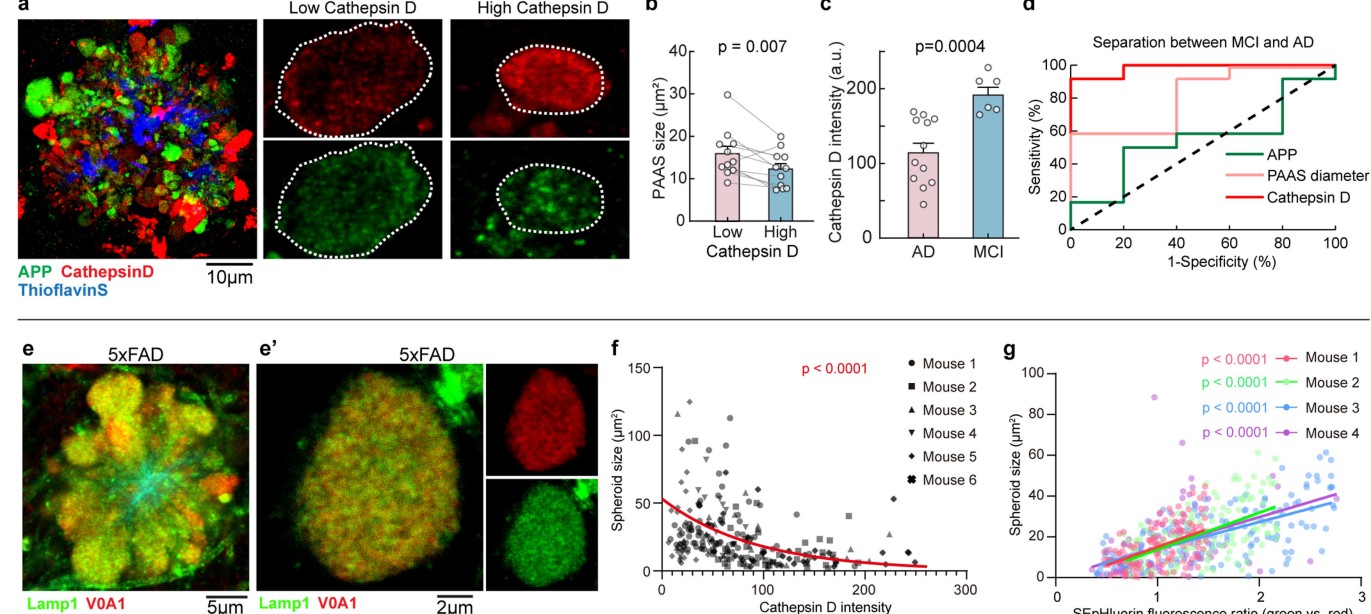

**Extended Data Fig. 6 | Additional data on the accumulation of enlarged vesicles within PAAS. a**, Confocal images of spheroids labelled by APP and Cathepsin D immunofluorescence in post-mortem human AD brain. Right panels show zoomed-in examples of spheroids (white dotted lines) with low or high Cathepsin D contents. All spheroids were positive with APP staining with variable labelling intensities. Some structures with saturated Cathepsin D labelling were likely to be microglia lysosomes. **b**, Spheroid size as a function of Cathepsin D contents. N = 11 human subjects. Each pair of dots represents average measurements from 50 spheroids in the same post-mortem brain. **c**, Comparisons of Cathepsin D levels within PAAS between AD and MCI patients. Each dot represents an average measurement of 50 PAAS. N = 12 AD and 6 MCI subjects. **d**, Receiver operating characteristic (ROC) curves clearly differentiate AD from MCI patients using PAAS diameter and APP or Cathepsin

D contents as parameters. **e-e'**, Confocal image showing colocalization of V0A1 and LAMP1 in axonal spheroids in 5xFAD mice. **f**, Scatter plot of the intensities of Cathepsin D immunofluorescence against spheroid sizes, related to Fig. 2f. Each dot represents an individual spheroid, with individual mice labelled with different shapes. N = 50 spheroids for each mouse. Red line shows an exponential regression from all data, and p-value shows Z-test of decay coefficient against zero. **g**, Scatter plot of the pH sensor green to red fluorescence ratios against spheroid sizes, related to Fig. 2h. Each dot represents an individual spheroid, and dot colour indicates individual mice. N = 150 spheroids for each mouse. Coloured lines show linear regressions for data from each mouse, and p-values show Z-test of slope being zero. Two-tailed paired t test was performed in **b**. Two-tailed Mann-Whitney test was performed in **c**. Data are represented as mean ± S.E.M.

**5xFAD mice**

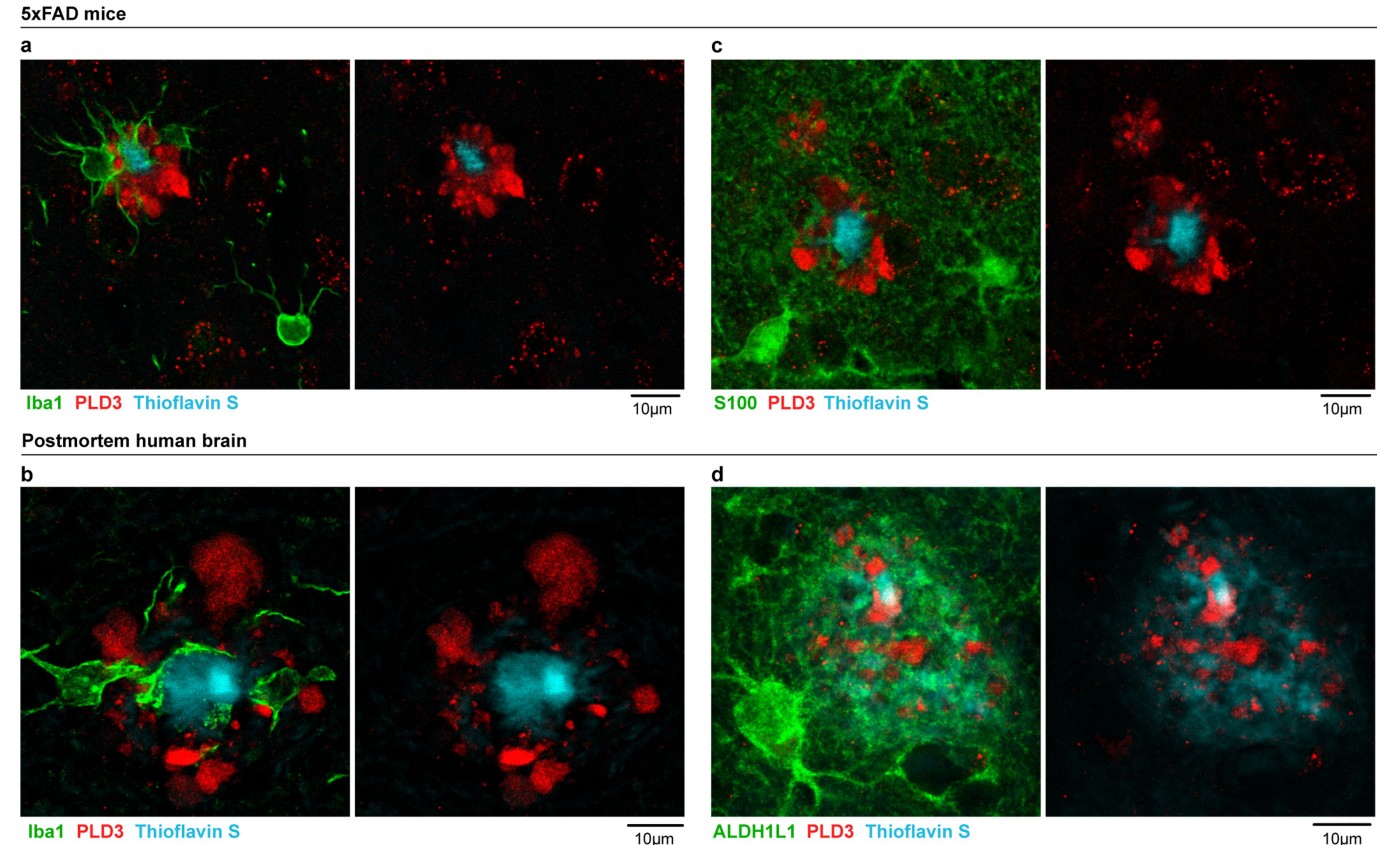

**Postmortem human brain**

**Extended Data Fig. 7 | Absence of PLD3 protein expression in microglia and astrocytes in 5xFAD mice or human AD brains. a** and **b**, Confocal images showing absence of PLD3 signal (red) within Iba1-labelled microglia (green) in 5xFAD mouse brain (**a**) and post-mortem brain tissue of AD human patients (**b**). **c**, Confocal imaging of 5xFAD mouse brain showing absence of PLD3 signal (red) within S100-labelled astrocytes (green). **d**, Confocal imaging of human AD post-mortem brain tissue showing absence of PLD3 signal (red) within ALDH1L1-labelled astrocytes (green).

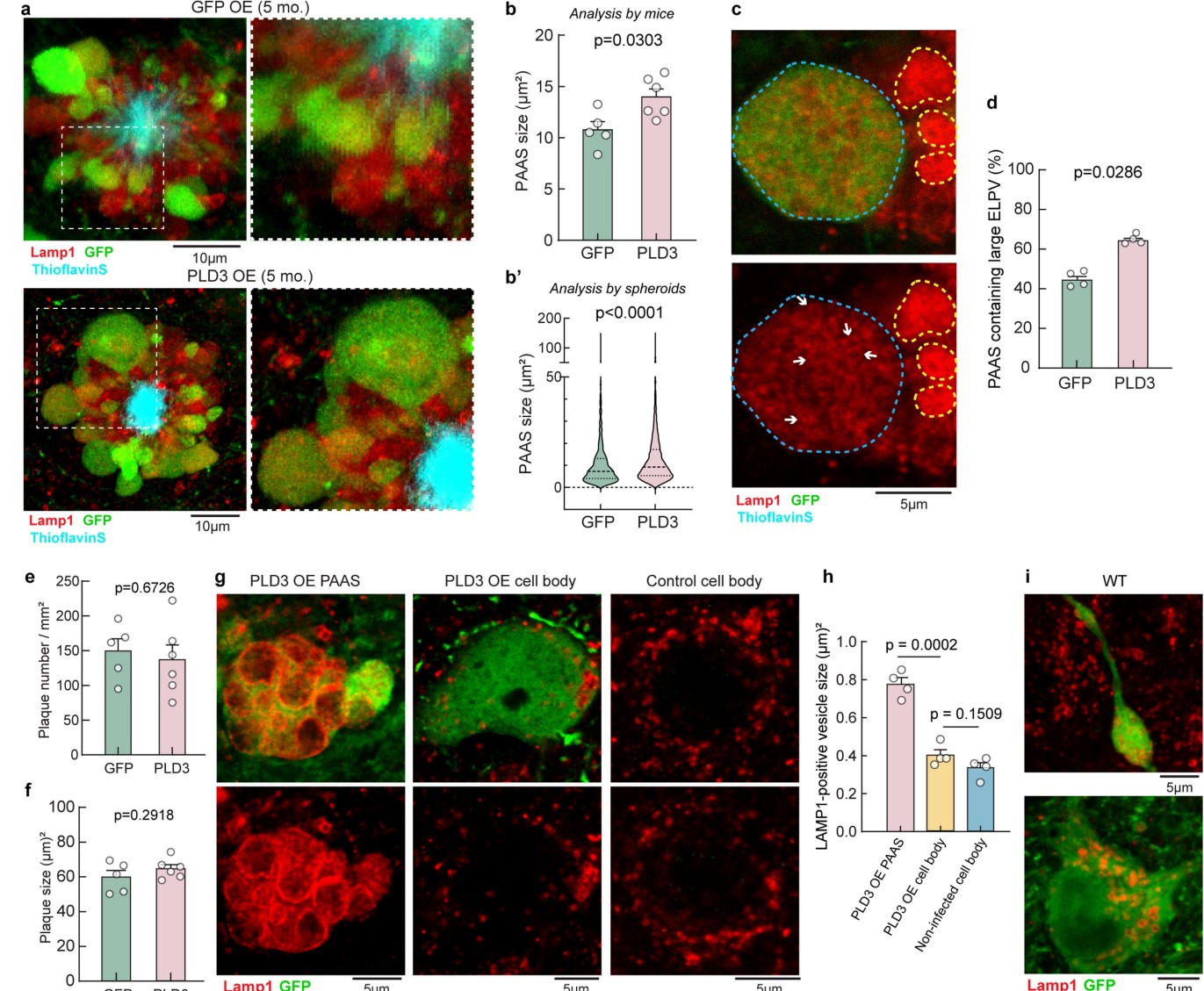

**Extended Data Fig. 8 | Analyses of ELPVs, spheroids and amyloid plaques in 5-month-old 5xFAD mice with PLD3 overexpression. a**, Confocal images of spheroids with GFP or PLD3 overexpression. Right panels show zoomed images from dashed boxes. **b**, Spheroid sizes in 5-month-old 5xFAD mice with PLD3 or GFP overexpression, presented by individual mice (**b**) or PAAS (**b'**). N = 6 and 5 mice for PLD3 and GFP groups, respectively; each dot represents average measurements from 350-600 PAAS. Violin plots show distributions of 1400-1600 individual PAAS from each group. **c**, Confocal images of adjacent PAAS with (blue dashed line) and without (yellow dashed line) PLD3 overexpression. Arrows indicated enlarged ELPVs. **d**, ELPV occurrence in PAAS in 5-month-old 5xFAD mice with PLD3 or GFP overexpression. N = 4 mice for each group. Each dot represents average measurements from 150-250 PAAS.

**e** and **f**, Plaque number (**e**) and size (**f**) in mice with GFP or PLD3 overexpression. N = 5 and 6 for GFP and PLD3 groups, respectively. For (**f**), each dot represents average measurements from 100-250 plaques. **g**, Confocal images of LAMP1-positive vesicular structures in PAAS and cell bodies in 10-month-old mice with PLD3 overexpression. **h**, ELPV sizes in groups described in **g**. N = 4 mice for each group. Each dot represents average measurements from 500-1000 ELPVs from PAAS or 100–200 LAMP1-positive vesicles from cell bodies. **i**, Confocal images of spheroids and LAMP1-positive vesicles in wildtype mice overexpressing PLD3. Two-tailed Mann-Whitney tests were performed in **b**, **b'** and **d**. Two-tailed unpaired t-tests were performed in **e** and **f**. Two-tailed Welch's t-test was used in **h**. Data are represented as mean ± S.E.M.

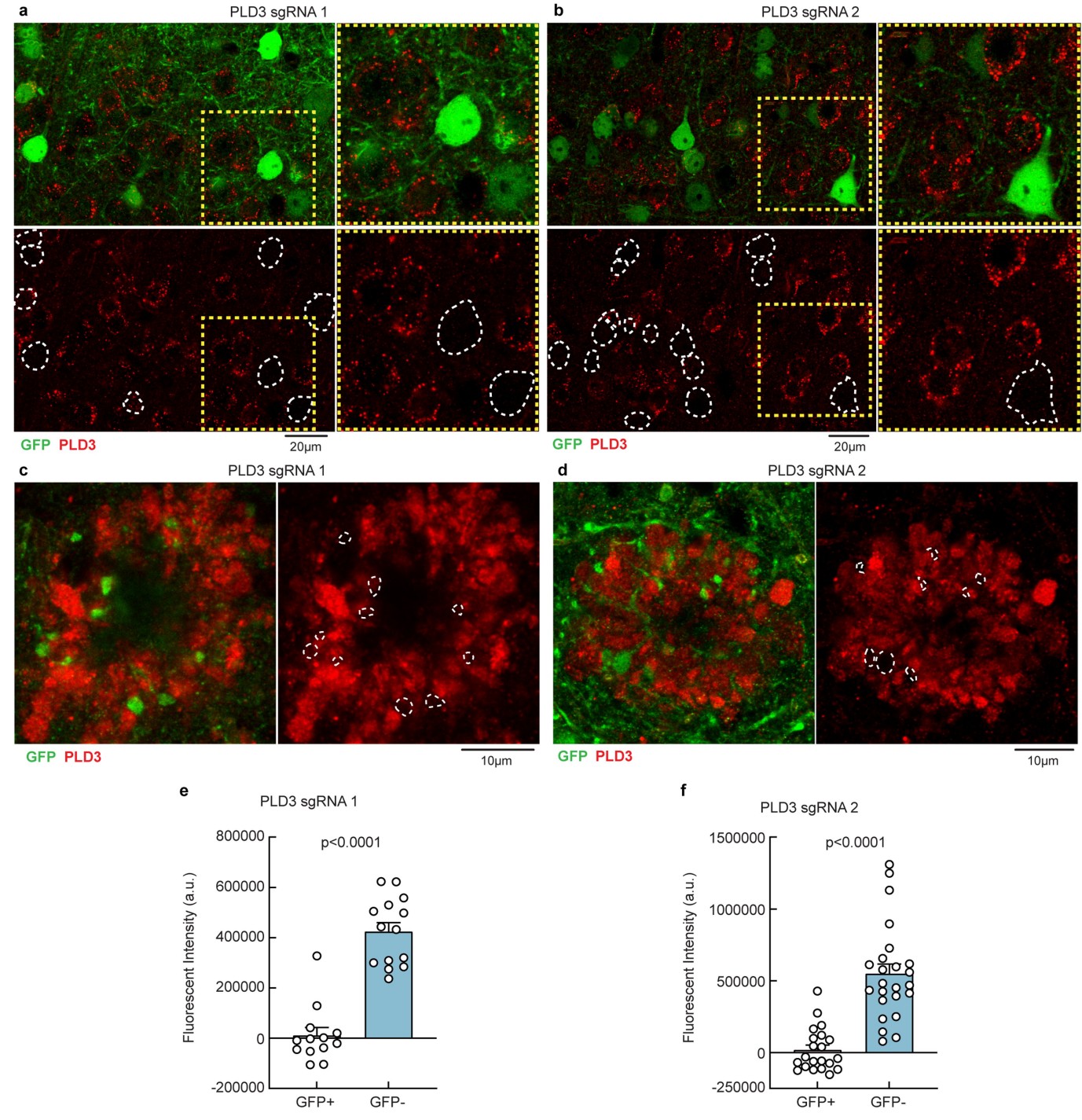

**Extended Data Fig. 9 | Validation of CRISPR-Cas9-mediated PLD3 deletion.**
**a-d**, Confocal images of PLD3 immunohistochemistry in tissue infected (green) and uninfected with PLD3-targeted sgRNA1 (**a** and **c**) or sgRNA2 (**b** and **d**). White dashed lines indicate outlines of infected cell bodies or individual spheroids. Yellow dashed lines indicate zoomed-in field of view on the right.

**e** and **f**, Quantifications of PLD3 fluorescence intensities following background subtraction in cell bodies with (GFP+) or without (GFP-) PLD3-targeted sgRNA1 (**e**) or sgRNA2 (**f**). Each dot represents average fluorescence of a cell body. N = 13 GFP+ and 14 GFP- cell bodies in **e**; N = 21 GFP+ and 24 GFP- cell bodies in **f**. Mann-Whitney tests were performed. Data are represented as mean ± S.E.M.

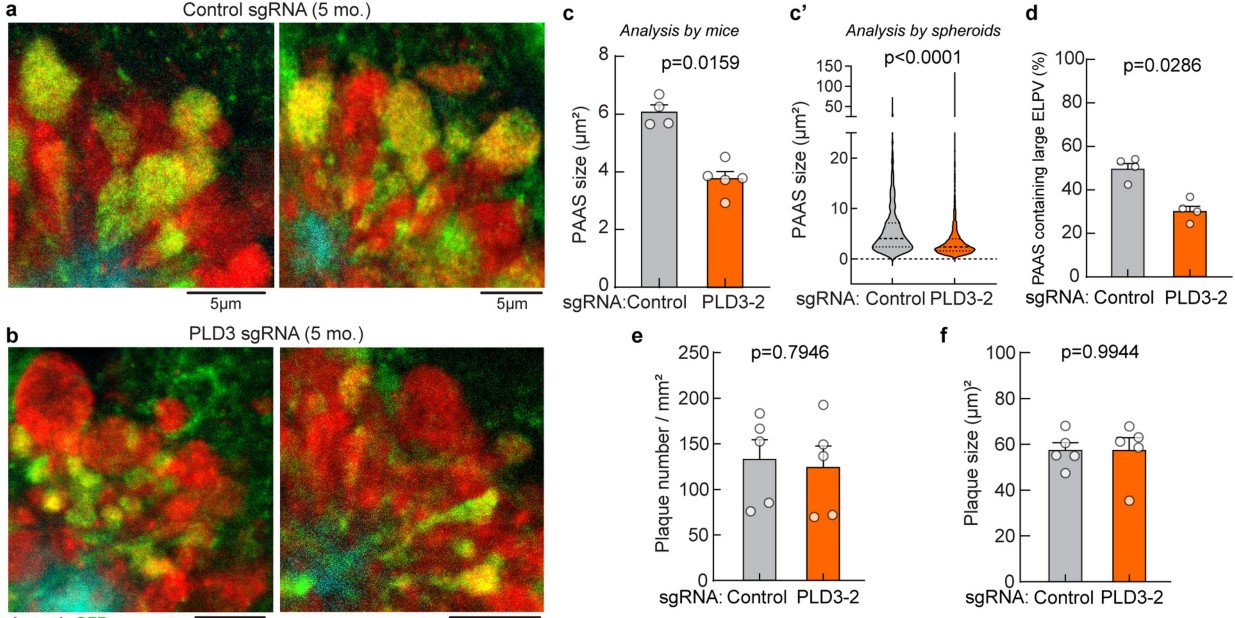

**Extended Data Fig. 10 | Additional analyses of ELPVs, spheroids and amyloid plaques in mice with PLD3 deletion. a** and **b**, Confocal images of infected and uninfected PAAS in 5xFAD mice with control sgRNA (**a**) or PLD3-targeting sgRNA (**b**). **c**, Spheroid size in 5-month-old mice with control sgRNA or PLD3 sgRNA 2, presented by individual mice (**c**) or PAAS (**c'**). N = 4 and 5 mice for control and PLD3 sgRNA groups, respectively; each dot represents average from 350-600 PAAS measurements. Violin plots show the distributions of 800-1200 individual PAAS from each group. **d**, ELPV occurrence in mice with control sgRNA or PLD3 sgRNA 2. N = 4 mice for each group. Each dot represents the average measurements of 150-250 PAAS. **e** and **f**, Plaque number (**e**) and size (**f**) in mice with control or PLD3 sgRNA. N = 5 mice for each group. Each dot represents average measurements of 100-250 plaques. Two-tailed Mann-Whitney tests were performed in **c**, **c'** and **d**. Two-tailed unpaired t-tests were performed in **e** and **f**. Data are represented as mean ± S.E.M.

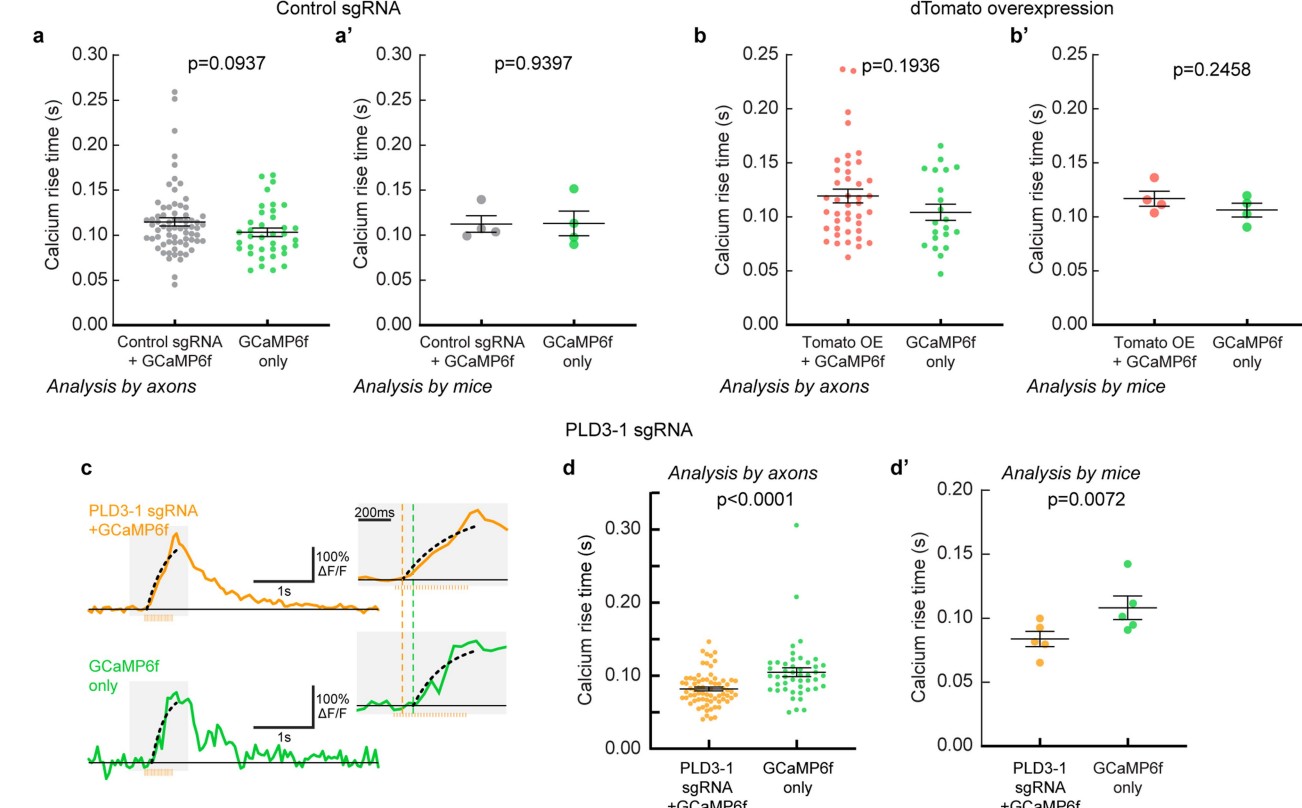

**Extended Data Fig. 11 | Additional analyses of axonal conduction upon PLD3 modulation. a**, Spike times in axons expressing control sgRNA, presented by individual axons or by mice. n = 69 manipulated and 37 control axons, and N = 4 mice. **b**, Spike times in axons with dTomato overexpression, presented by individual axons (**b**) or by mice (**b'**). n = 42 manipulated and 21 control axons, from N = 4 mice. **c**, Example traces of calcium dynamics in contralateral axons following PLD3 deletion with sgRNA-1. Yellow flash icon indicates the time of stimulation. Orange bars show the 50 Hz spike train for stimulation. The inserts show zoomed-in plots of the calcium transients (grey rectangles). Black lines indicate exponential regressions of the rising phase and coloured vertical dashed lines show estimated spike times. **d**, Spike times in axons with PLD3-deletion using sgRNA-1, presented by individual axons (**d**) or by mice (**d'**). n = 71 manipulated and 46 control axons, and N = 5 mice. Two-tailed Mann-Whitney tests were performed in **a**, **b** and **d**. Two-tailed paired t-test were performed in **a'**, **b'** and **d'**. Data are represented as mean ± S.E.M.

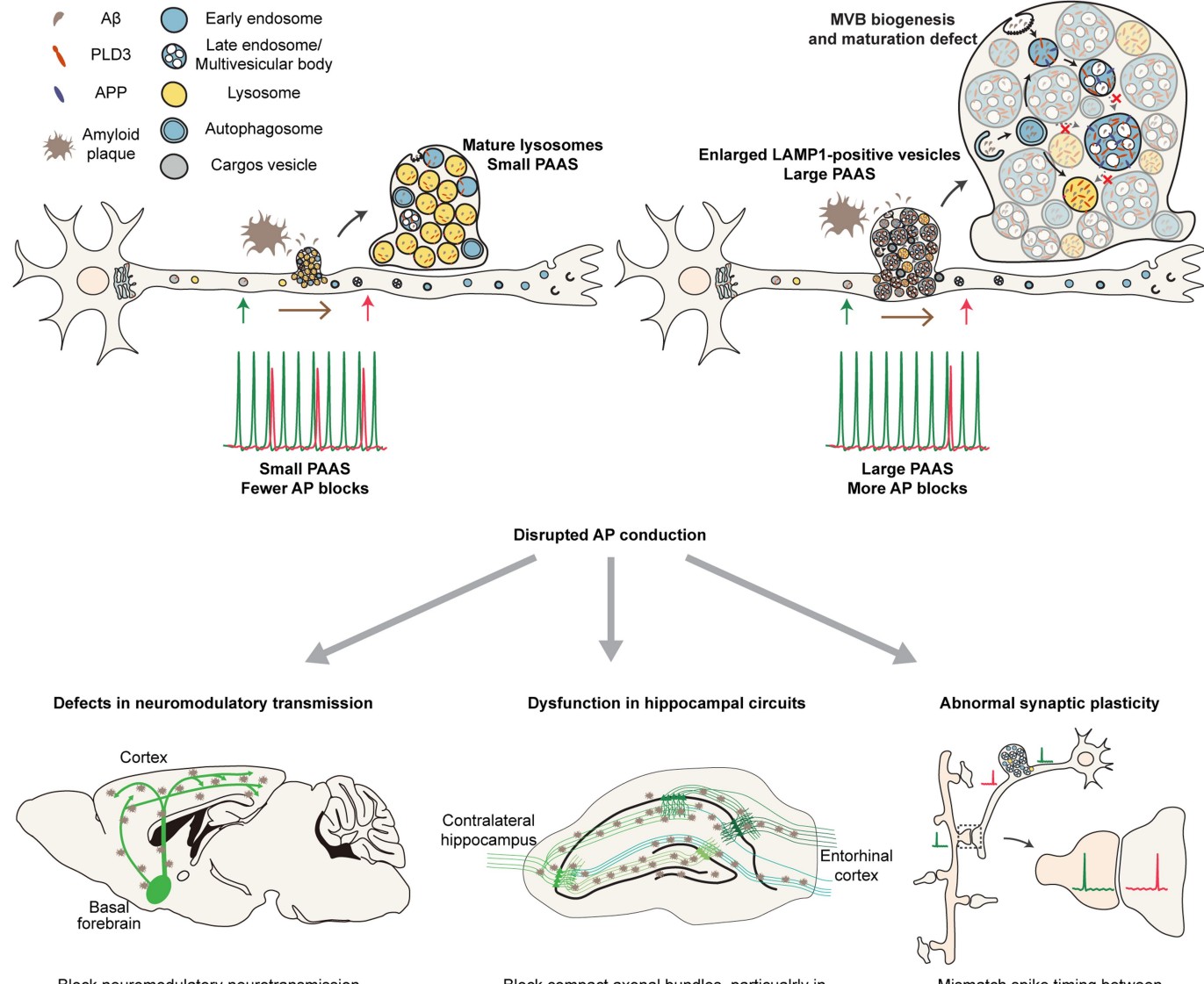

**Extended Data Fig. 12 | Proposed model of PAAS enlargement and functional consequences in Alzheimer's disease.** 1) Our study demonstrated that the accumulation of abnormally enlarged ELPVs is a major driver of PAAS enlargement. Small PAAS predominately contain mature lysosomes, while larger PAAS contain abundant and enlarged ELPVs. 2) We identified PLD3 as a critical modulator of MVB abnormalities and subsequent spheroid enlargement. PLD3 is uniquely sorted through the ESCRT pathway into the intralumenal vesicles (ILVs) of MVBs. The accumulation of PLD3 in spheroids could lead to MVB enlargement by interfering with ESCRT machinery. This process could be exacerbated with the presence of Aβ. Aβ from extracellular amyloid deposits is actively endocytosed and is present in the same subcellular compartments as PLD3. PLD3 could thus work synergistically with Aβ, leading to greater MVB abnormalities. 3) Large PAAS cause more severe conduction blocks, by functioning as current sinks. Given that hundreds of axons around each plaque develop spheroids and these structures remain stable for extended periods of times, the large number of plaques present in the AD brain could significantly affect neural networks by widespread disruption of axonal connectivity. 4) We found that cortical neurons in 5xFAD mice exhibited hyperactivity, and this can be corrected by restoring axon conduction through reducing PAAS in basal forebrain cholinergic projections. This suggests that PAAS can cause widespread disruption of neural circuit function. In addition, parallel compact axonal bundles that follow a stereotyped projection path along a tri-synaptic loop in hippocampus, a region critical for memory formation, could be particularly vulnerable to amyloid plaques located in the region. Furthermore, neural processes that rely on temporally precise long-range coordination among brain regions, such as memory consolidation, could be severely affected. In addition, synaptic plasticity could also be disrupted, due to the requirement of precise timing of firing between presynaptic and postsynaptic terminals. Altogether, action potential blocks caused by PAAS could be detrimental to various neural processes such as memory formation and reaction time, potentially contributing to cognitive decline in AD.

# Reporting Summary

## Statistics

For all statistical analyses, confirm that the following items are present in the figure legend, table legend, main text, or Methods section.

| n/a | Confirmed | |
|---|---|---|
| ☐ | ☒ | The exact sample size (*n*) for each experimental group/condition, given as a discrete number and unit of measurement |
| ☐ | ☒ | A statement on whether measurements were taken from distinct samples or whether the same sample was measured repeatedly |
| ☐ | ☒ | The statistical test(s) used AND whether they are one- or two-sided *Only common tests should be described solely by name; describe more complex techniques in the Methods section.* |
| ☐ | ☒ | A description of all covariates tested |
| ☐ | ☒ | A description of any assumptions or corrections, such as tests of normality and adjustment for multiple comparisons |
| ☐ | ☒ | A full description of the statistical parameters including central tendency (e.g. means) or other basic estimates (e.g. regression coefficient) AND variation (e.g. standard deviation) or associated estimates of uncertainty (e.g. confidence intervals) |
| ☐ | ☒ | For null hypothesis testing, the test statistic (e.g. *F*, *t*, *r*) with confidence intervals, effect sizes, degrees of freedom and *P* value noted *Give P values as exact values whenever suitable.* |
| ☒ | ☐ | For Bayesian analysis, information on the choice of priors and Markov chain Monte Carlo settings |
| ☒ | ☐ | For hierarchical and complex designs, identification of the appropriate level for tests and full reporting of outcomes |
| ☐ | ☒ | Estimates of effect sizes (e.g. Cohen's *d*, Pearson's *r*), indicating how they were calculated |

*Our web collection on statistics for biologists contains articles on many of the points above.*

## Software and code

Policy information about availability of computer code

| Data collection | Confocal images were collected using a Leica SP5 or SP8 system with the associated software. Two-photon images were collected using a Prairie Technology system using the associated Prairie View software (version 5.4), or the Bruker Ultima Investigator multi-photon microscope system. Computational modeling of action potential propagation was carried out in the NEURON environment. |
|---|---|
| Data analysis | Images were analyzed using customized codes in NIH FIJI and MATLAB software. Statistics were performed with GraphPad Prism (version 7, 8 and 9) software. All codes are available online: https://github.com/PaulYJ/Axon-spheroid. |

For manuscripts utilizing custom algorithms or software that are central to the research but not yet described in published literature, software must be made available to editors and reviewers. We strongly encourage code deposition in a community repository (e.g. GitHub). See the Nature Portfolio guidelines for submitting code & software for further information.

## Data

Policy information about availability of data

All manuscripts must include a data availability statement. This statement should provide the following information, where applicable:
- Accession codes, unique identifiers, or web links for publicly available datasets
- A description of any restrictions on data availability
- For clinical datasets or third party data, please ensure that the statement adheres to our policy

The authors declare that all derived data supporting the findings of this study are available within the paper and its supplementary information files.

March 2021

# Human research participants

Policy information about studies involving human research participants and Sex and Gender in Research.

| Reporting on sex and gender | We obtained postmortem fixed brain tissue from 16 patients with Alzheimer's disease (8 male and 8 female) and 6 with mild cognitive impairment (3 male and 3 female). We did not observe sex-specific effect in our results. |
|---|---|
| Population characteristics | The detailed demographic information of all the subjects can be found in extended data figure 4. In short we tried our best to balance the sample's age, gender, and ApoE genotype in each group. |
| Recruitment | The brain tissues were obtained from established brain banks, with standard procedure of informed consent for body donation. |
| Ethics oversight | The brain tissues were obtained from the Sun Health Research Institute Brain and Body Donation Program, the Mayo Clinic , the University of Washington Alzheimer's Disease Research Center Neuropathology Core and the Alzheimer's Disease Research Center at Washington University at St. Louis (The Knight ADRC Biospecimens Committee). All in complete compliance with the human tissue research use regulation in each organization. |

Note that full information on the approval of the study protocol must also be provided in the manuscript.

# Field-specific reporting

Please select the one below that is the best fit for your research. If you are not sure, read the appropriate sections before making your selection.

☒ Life sciences　　☐ Behavioural & social sciences　　☐ Ecological, evolutionary & environmental sciences

For a reference copy of the document with all sections, see nature.com/documents/nr-reporting-summary-flat.pdf

# Life sciences study design

All studies must disclose on these points even when the disclosure is negative.

| Sample size | For PAAS treatment experiment, the sample size was determined based on previous experiments measuring this pathology (Neuron 90, 724–739, 2016).<br>For calcium imaging of single axons, the sample size was not pre-determined due to the unpredictability of the imaged spheroids. The sample size is justified since the variance within group is small compared to between group differences.<br>For calcium and voltage imaging of interhemispheric axon conduction, a large number of axons were imaged in multiple mice for each group. The observation was highly consistent between mice within each group, indicating that the sample size used was sufficient. |
|---|---|
| Data exclusions | No data was excluded from the study. |
| Replication | Three key results of the main paper were replicated.<br>1, PLD3 overexpression was independently repeated once with a different batch of mice.<br>2, PLD3 deletion was independently repeated once with a different batch of mice.<br>3, In vivo calcium imaging has been independently performed by 3 different experimenters, each experimenter has imaged multiple mice from 2019 to 2021. In vivo voltage imaging has been indenpendently performed by 2 different experimenters, each experiment has imaged multiple mice from late 2021 to early 2022.<br>All replications successfully reproduced reported results. |
| Randomization | In all experiment related to the treatment of PAAS (including quantification of spheroids and calcium/voltage recovery experiments), we used age-matched mice, and then randomly administer the mice with treatment virus or control virus. Sex balance were also taken into consideration. |
| Blinding | In all experiment related to the treatment of PAAS, images from treatment and control groups were blinded by replacing the file names to random identifiers during analysis via a custom FIJI script. The treatment assignment was only revealed after the measurement of all axon spheroids from the whole experiment was completed. |

# Reporting for specific materials, systems and methods

We require information from authors about some types of materials, experimental systems and methods used in many studies. Here, indicate whether each material, system or method listed is relevant to your study. If you are not sure if a list item applies to your research, read the appropriate section before selecting a response.

## Materials & experimental systems

| n/a | Involved in the study |
|---|---|
| ☐ | ☒ Antibodies |
| ☐ | ☒ Eukaryotic cell lines |
| ☒ | ☐ Palaeontology and archaeology |
| ☐ | ☒ Animals and other organisms |
| ☒ | ☐ Clinical data |
| ☒ | ☐ Dual use research of concern |

## Methods

| n/a | Involved in the study |
|---|---|
| ☒ | ☐ ChIP-seq |
| ☒ | ☐ Flow cytometry |
| ☒ | ☐ MRI-based neuroimaging |

# Antibodies

| Antibodies used | 1. anti-LAMP1 (DSHB, 1D4B),<br>2. anti-GFP (Aves Labs. Inc. GFP-1020),<br>3. anti-CathepsinD (Abcam, EPR3057Y, ab75852),<br>4. anti-ATP6V0A1 (ThermoFisher Scientific, PA5-54570),<br>5. anti-amyloid precursor protein (ThermoFisher Scientific, LN27, 13-0200),<br>6. anti-PLD3 (Sigma-Aldrich, HPA012800),<br>7. anti-beta amyloid 1-42 (Abcam, ab10148),<br>8. anti-beta amyloid 1-42 (Abcam, mOC98, ab201061),<br>9. anti-MAP2 (Abcam, ab5392),<br>10. anti-Iba1 (Novus Biologicals, NB100-1028),<br>11. anti-S100B (R&D Systems, AF1820),<br>12. anti-Aldh1l1 (NeuroMab, P28037).<br>13. Alexa Fluor dye conjugated secondary antibodies from ThermoFisher. |
|---|---|
| Validation | All antibodies have been validated in lab through immunofluorescence, to confirm the staining pattern is consistent with that shown on the manufacturer website or by previous literature. In addition, the antibodies have also been validated by the manufacturer as shown by western blot and/or immunofluorescence images on their website. |

# Eukaryotic cell lines

Policy information about cell lines and Sex and Gender in Research

| Cell line source(s) | HEK293T: American Type Culture Collection (ATCC) |
|---|---|
| Authentication | Authentication was guaranteed by the supplier and authors did not carry out any authentication of the cell lines. The HEK293T cells was used for AAV production only, not for experiments. |
| Mycoplasma contamination | Cell lines were not tested for mycoplasma. |
| Commonly misidentified lines (See ICLAC register) | None. |

# Animals and other research organisms

Policy information about studies involving animals; ARRIVE guidelines recommended for reporting animal research, and Sex and Gender in Research

| Laboratory animals | 5xFAD (34840-JAX, The Jackson Laboratory) mice were used in this study. Rosa26-LSL-Cas9 (026175, The Jackson Laboratory) mice were crossed with 5xFAD mice for CRSIPR/Cas9-mediated gene deletion. |
|---|---|
| Wild animals | None. |
| Reporting on sex | We balanced both sexes in all our experiments to the best of our capability. No sex-specific effects of axonal spheroids on conduction, or modulation of PLD3 on the degree of axonal spheroids were observed in our study. |
| Field-collected samples | None. |
| Ethics oversight | Institutional Animal Care & Use Committee (IACUC) at Yale University. |

Note that full information on the approval of the study protocol must also be provided in the manuscript.

nature portfolio | reporting summary

March 2021

