## [Peer Review File · Nature]

Manuscript Title: Axonal and network defects in Alzheimer's model reversed by PLD3 modulation.

Reviewer Comments & Author Rebuttals

Reviewer Reports on the Initial Version:

Referees' comments:

Referee #1 (Remarks to the Author):

This is a very interesting study that links the presence of axonal swellings (spheroids) to axonal conductance problems. The authors imply the intraluminal protein PLD3 in the generation of these spheroids and show that Abeta42 is endocytosed in the intraluminal vesicles of MVB which constitute the major structural element of the spheroids.

The work is well done and expands our knowledge of a phenomenon that has been noticed regularly but has been little investigated, apart from morphological description. It links amyloid (plaque) pathology to functional disturbances in axonal guidance and links a (potential) genetic risk factor of AD to a pathological effect in AD and mechanisms of endocytosis/lysosomal biology/autophagy which are of high interest in the AD field. The paper provides a consistent explanation on how amyloid plaques could cause cognitive alterations in AD. The authors support their model with additional circumstantial evidence in human AD brain, strengthening the relevance of their findings for human disease.

While the work, admittedly, is breaking ground and very exciting for the field, it is nevertheless clear that not all links in this story are fully explored and clarified. Obviously it is unreasonable to expect that all issues are solved in a first publication, and additional investigation by the community will certainly explore this story further. I wonder therefore to what extent more precise wording can accommodate or clarify this, and to what extent additional experimentation should be provided to support more strongly causality. This is an editorial decision.

Specific criticism

1. Some of the conclusions remain correlative and need further investigation. The authors claim for example that the axonal spheroids are induced by the presence of amyloid plaques and that they correlate with the size of the plaques. This is based on (limited) morphological correlation. Can the authors reverse the amount of spheroids by reducing amyloid plaques in these mice? What is the mechanism that induces these spheroids? Can they induce spheroids in vitro by adding Abeta42 to neurons, for instance in combination with PLD3 overexpression?

R118-121: is speculative.

R140-141: the spheroid enlargement correlates with accumulation of MVB, the mechanistically link suggested in r140 is however not proven.

R147-r149: again correlation does not implicate causality.

2. It is unclear how spheroids are defined in human brain, see for instance my questions regarding fig2k, below. How are they identified and how are their surfaces calculated?

3. The authors ignore in their article that PLD3 is associated to AD in a loss of function scenario by the original study (Cruchaga et al). Thus it should be discussed why the authors think that the

previous interpretation was wrong. Do the authors have evidence for upregulation of PLD3 in AD?
4. The authors speculate about a direct interaction between PLD3 and Abeta. This needs further clarification.

Minor remarks

1. R175: ref 28 Fazarri et al provided evidence from in vivo experiments, the work encompassed crosses of PLD3 ko with APP knock in mice.
2. Fig 1 d needs some more explanation. What is the green spot in the circle? What are the green cells depicted on top of the corpus callosum? I do not understand what happens: it seems that the cell bodies in the contralateral side are stimulated, but how is then t1 and t2 determined, i.e. how do you know that the signal in t1 is upstream of the spheroid and in t2 downstream?
3. Fig 1 e and e' : what is depicted in the Y-axis of the panels?
4. Fig 1g AS is used as abbreviation in legend, do the authors mean PAAS? AS is axonal segment and also used as antisense in other parts of the MS.
5. Fig 1g Is the number of axons measured not very small (n=10, 21 and 8 and are obtained from 14 mice, implying that <3 axons per mice were checked)
6. Fig 2c: it is unclear how the panel has been generated. As the dots represent means of measurements, I do not understand what the coloured bars mean. From these data it seems that all PAAS have very similar sizes (60 μm^2)? It might be more meaningful if for each spot the mean +/-SD is given so that we get an understanding of the spreading in those data points
7. Fig 2k: the panel shows cathepsin d and APP staining. I suppose that the APP staining reflects here the spheroids, but then there is clearly no colocalisation with cathepsin D, or vice versa and how does this picture relate to the panels at the right?
8. Fig 3c and c' : not indicated which area is zoomed in. I suppose that the GFP overexpression is coupled to GFP overexpression?

Referee #2 (Remarks to the Author):

This is a peer review for a new manuscript by Prof Grutzendler and colleagues wherein they describe the impact of dystrophic neurites on action potential conductance. They discovered that dystrophic neurites slow the conductance of action potentials and that this relationship is related to the size of dystrophic neurites, larger neurites producing greater slowing or even blocking conductance. They describe the presence of multivesicular bodies in dystrophic neurites, which they postulate are associated with larger size of the neurites. They report that phospholipase D3, a variant of which is reportedly a genetic risk factor for AD, is associated with these large multivesicular bodies. Overexpression of PLD3 via neuronally targeted expression driven by a viral vector resulted in expansion of dystrophic neurites while CRISPR-Cas9 mediated knock-down resulted in smaller dystrophic neurites. As with previous work from this group, the quality of the live-imaging microscopy is exceptional and creates a lot of visual appeal for the manuscript. Quantitative methodologies are used throughout the work and, while I am not a statistician, the statistical approaches seem reasonable. In a couple of instances specifically noted below, the experiments had too small of a sample size.

Figure 1: Establishing the electrophysiologic significance of dystrophic neurites has implications for connectivity and potentially for cognition, thus it is a priority research topic. Using live calcium

imaging to visualize action potential conductance across dystrophic neurites is novel and from this data the authors conclude that dystrophic neurites act as capacitors to slow (or block) action potential conductance. Importantly, calcium transients are slower than action potential generation/conduction within axons and some calcium channels are low-threshold channels which may not need action potentials to be activated. Consequently, while slowed action potential conductance across dystrophic neurites is a reasonable hypothesis, I am not sure the claim can be robustly supported using the calcium imaging method. Direct electrophysiological measurement or use of a voltage sensitive dye or protein may provide informative alternatives.

The modeling of the extent of axonal dystrophy associated with each plaque is of interest. I am not sure I agree with all of the assumptions the authors use in this analysis (notably, unlike the 5xFAD model, many plaques in human brain have no dystrophic neurites at all), but the calculations are probably useful to at least provide a sense of the scale of the process.

The conclusion that dystrophic neurites are a modifiable pathology rather than degenerative based on long-term live imaging in 5xFAD mice is somewhat overstated. The 5xFAD mouse does not model neurodegeneration or mature tau pathology. Dystrophic neurites in human tissue often contain tau aggregates and likely contain a different balance of organelles than the 5xFAD mouse – it remains quite plausible that the equivalent process in human brain is less reversible. I would moderate the interpretation of this result, but never-the-less, the long-term microscopy of these structures is an accomplishment of considerable interest.

Figure 2: I am not convinced that the large LAMP1-positive vacuoles that are seen in the immunohistochemistry of dystrophic neurites represent the same type of organelle as that shown in the electron microscopy. The vacuoles on IHC seem quite a lot larger than the representative MVB on the EM. There are also other organelles adjacent to the MVB on the representative EM image that are of similar size. Conceptually, quite a few organelles could be candidates for a LAMP+ vacuole, including pathologically enlarged lysosomes or organelles in the later stages of the autophagic pathway. I also think it is important to verify the result in human tissue as there is some discrepancy in the descriptions in prior work of the types of organelles in dystrophic neurites between animal models and human neuropathological studies.

I found the connection between the extent of axonal dystrophy pathology and cognition to be tenuous. The analysis of human tissue included no control subjects, 6 subjects with mild cognitive impairment at advanced ages and 12 subjects with AD over a range of ages. Aside from the categorical characterization, there was no information on their cognitive status. MCI, of course, represents a range of underlying processes (including in many cases normal aging) and it is possible that non-amyloid pathologies contributed. The use of APP as the marker of dystrophic neurites in this setting is questionable (fig2k shows the labeling with APP in human brain inconsistently marked dystrophic neurites). I would have preferred the authors utilized existing, larger-scale patient cohorts with neuropathological data to establish a rigorous association between dystrophic neurites with cognition.

Figure 3: I do not find the evidence linking the staining pattern of PLD3 in dystrophic neurites to MVB convincing. PLD3 seems to be distributed throughout the dystrophic neurite, with some membrane associated and some not, as others have previously reported. The suggestion that PLD3 is linked to the biogenesis of MVBs is not supported by this data and does not consider the possibility of failed maturation or trafficking defects.

I did find the panels linking PLD3 expression to the size of dystrophic neurites compelling (panels C-H), although in prior figures the analysis was shown both per-plaque and per-mouse; that pattern should be continued. We don't really understand what factors are central to dystrophic neurite formation and growth, so this result is informative.

The final panels in this figure (I-N) are under-developed and may be unnecessary. In a mouse model with non-physiologic levels of over-production of β -amyloid, it is not surprising that β -amyloid is present in neuronal lysosomes and certainly does not strongly support the notion that it was endocytosed. This experiment could have been performed in human tissue with a more convincing result. The use of Dynasore in the subsequent experiment is not discussed in the text or methods at all, there are no methods given for the slice culture, and the interpretation of the result is underdeveloped. In the final panels, the preparation of the β -amyloid for injection is not described, the number of replicates is small, no controls are used and there are a number of confounders and alternative explanations for these experiments which are not discussed.

Figure 4: Silencing PLD3 expression reduced the size of dystrophic neurites. This is a striking finding. Importantly, the demonstration that PLD3 knock-down in mouse neurons resulted in reduced size of dystrophic neurites set up the opportunity to causally demonstrate a link between dystrophic neurites and cognition through neurobehavioral testing. Cognitive data would more convincingly demonstrate that PLD3 is a reasonable drug target and would strengthen this work.

Cumulatively, I found the evidence that PLD3 regulates the size of dystrophic neurites convincing and potentially of considerable importance. Strengthening the link between PLD3 in dystrophic neurites and cognition would greatly enhance the study. The conclusion that vacuoles in dystrophic neurites are multivesicular bodies and that dystrophic neurites impair action potential conductance cannot be conclusively made from the available data, in my opinion.

Referee #3 (Remarks to the Author):

The manuscript identifies plaque-associated axonal spheroids (PAAS) as contributors to action potential conduction delays and blockades in AD mice using calcium imaging of single axons in vivo. PAAS were stable over time and larger size was associated with increased conduction deficits. PAAS accumulated large multivesicular bodies (MVBs) and PLD3, a lysosomal protein. Mechanistically, overexpression of PLD3 led to accumulation of MVBs, PAAS enlargement and action potential conduction deficits, whereas PLD3 depletions reduced these deficits. Overall, the manuscript suggests that PLD3-dependent PAAS may contribute to network dysfunction in AD.

The manuscript addresses functionally for the first time and in a rigorous way one of the hallmarks of AD pathology, axonal dystrophic neurites, and identifies PLD3 as a potential mechanism of PAAS-induced action potential alterations. The experimental design is rigorous and elegant assessing single axon function with contralateral injections of tracers. Overall, the main conclusions are well supported by the experimental data, and results are properly discussed. I did not identify major deficiencies, but I have a few comments:

1) The authors repeatedly argue that PAAS contribute to “neural network dysfunction”. However, there is no assessment of neural network dysfunction by electrophysiological (e.g., EEG/LFP recordings) or behavioral approaches. GCaMP6f imaging before and after PAAS in single axons is not a measure of “neural network dysfunction”. There is only assessment at the axonal level and it seems too speculative to link these findings to network functions.

2) The overall conclusion that PLD3-dependent PAAS contribute to network dysfunction is weakly supported since no measures of network dysfunction are assessed in 5xFAD mice after PLD3 deletion.

3) Mouse PAAS are identified with a LAMP1-positive multivesicular bodies (MVBs) antibody, whereas human PAAS are labeled with APP or V0A1 (ATPase H transporting V0 subunit A1) antibodies, suggesting that human and mouse PAAS are not mechanistically equivalent. The authors also use the term “MVB-like structures” for human PAAS. Are mouse and human PAAS molecularly distinct?
Minor:

1) The term “action potential spike time” is not accurate since calcium signal relates to trains of action potential during the 500-ms stimulation. Calcium rise time would be more appropriate for describing the data.

2) Inserts in calcium figures do not contain temporal scale so it is difficult to assess the time.

3) Electrical stimulation is marked by an arrow giving the impression that is very brief, but these stimulations are very long for action potential dynamics (500 ms). Please indicate with a line the 500 ms of stimulation to properly assess the calcium data.

4) Fig 4g, calcium signal is observed before the stimulation, why?

5) The term “plaque-associated axonal spheroids” needs some clarification, since PAAS are also observed far from plaques (e.g., extended figure 4).

Author Rebuttals to Initial Comments:

Referee #1 (Remarks to the Author):

This is a very interesting study that links the presence of axonal swellings (spheroids) to axonal conductance problems. The authors imply the intraluminal protein PLD3 in the generation of these spheroids and show that Abeta42 is endocytosed in the intraluminal vesicles of MVB which constitute the major structural element of the spheroids. The work is well done and expands our knowledge of a phenomenon that has been noticed regularly but has been little investigated, apart from morphological description. It links amyloid (plaque) pathology to functional disturbances in axonal guidance and links a (potential) genetic risk factor of AD to a pathological effect in AD and mechanisms of endocytosis/lysosomal biology/autophagy which are of high interest in the AD field. The paper provides a consistent explanation on how amyloid plaques could cause cognitive alterations in AD. The authors support their model with additional circumstantial evidence in human AD brain, strengthening the relevance of their findings for human disease. While the work, admittedly, is breaking ground and very exciting for the field, it is nevertheless clear that not all links in this story are fully explored and clarified. Obviously, it is unreasonable to expect that all issues are solved in a first publication, and additional investigation by the community will certainly explore this story further. I wonder therefore to what extent more precise wording can accommodate or clarify this, and to what extent additional experimentation should be provided to support more strongly causality. This is an editorial decision.

Specific criticism

1. Some of the conclusions remain correlative and need further investigation. The authors claim for example that the axonal spheroids are induced by the presence of amyloid plaques and that they correlate with the size of the plaques. This is based on (limited) morphological correlation. Can the authors reverse the amount of spheroids by reducing amyloid plaques in these mice? What is the mechanism that induces these spheroids? Can they induce spheroids in vitro by adding Abeta42 to neurons, for instance in combination with PLD3 overexpression?

The reviewer asked an excellent question regarding the biological mechanism of spheroid formation, which still requires a lot of detailed investigation. With regards to the reversibility by removal of A β in mice, we showed in a previous publication that this reduced spheroids moderately (PMID: 25630253), but this is complicated by the fact that anti-Abeta antibody treatments induce significant glial activation and changes in the plaque microenvironment. We have done extensive work in the past showing the critical role of microglia in protecting adjacent axons through a barrier mechanism (PMID: 25630253 and 27196974). The presence of this mechanism complicates any experiment trying to assess the role of plaque removal on spheroids. However, recent human induced pluripotent stem cell (iPSC) data, clearly shows that spheroids that look almost identical to those in humans can be recapitulated in vitro by administration of aggregated A β (PMID: 34471104), thus establishing possible causation. Interestingly, these spheroids only form at sites of aggregated compact amyloid rather than adjacent diffuse deposits, consistent with previous observations in humans including our own (PMID: 27196974).

Regarding the full mechanism of spheroid formation, we agree with the reviewer that this is a very important question. While we do not know the full details of the process, our work identified an endogenous neuronal molecule that modulates this process. More research is required to understand the biochemical events that link A β , PLD3 and endolysosomal biogenesis leading to spheroid growth. It will also be critical to identify the early events that trigger the initial process of spheroid formation. We believe that there will be a surge of interest following the current study for delineating the early mechanisms controlling the formation and growth of this potentially important AD pathological hallmark.

R118-121: is speculative.

Here we hypothesized that long axons have a greater probability of encountering amyloid plaques at some point through their anatomy. It would be nice to show this with experimental data, however, tracing such axons throughout the brain is a very difficult task. Given these challenges, we would like to leave the statement as it currently is, because

we do think it is reasonable to speculate that axonal length determines the likelihood of interaction with extracellular amyloid deposits.

R140-141: the spheroid enlargement correlates with accumulation of MVB, the mechanistically link suggested in r140 is however not proven. R147-r149: again correlation does not implicate causality.

We agree with the reviewer that establishing ultimate causation is very difficult. However, our cumulative data does support the idea that alteration in endolysosomal signaling, through accumulation or overexpression of PLD3 is associated with formation of aberrantly large endolysosomal vesicles (including multivesicular bodies) and that this is strongly correlated with spheroid size. In contrast, deletion of PLD3 has the opposite effect. While we have not officially proven beyond doubt that the accumulation of these large organelles is what ultimately drives spheroid enlargement, we do think that this is a reasonable conclusion with the evidence at hand.

2. It is unclear how spheroids are defined in human brain, see for instance my questions regarding fig2k, below. How are they identified and how are their surfaces calculated?

We thank the reviewer for this question. Historically, the spheroids were labeled in human brain with anti-APP immunostaining (PMID: 9829809). In practice, as the reviewer pointed out, some spheroids show very low staining of APP, although distinguishable from the background. In our study, we have explored multiple markers for spheroids. We identified ATPase H transporting VO subunit A1(VOA1) as the most robust marker (Figure 2 and Extended figure 6). Furthermore, when comparing the number of spheroids across different groups in humans, we repeated the quantification using the anti-APP and anti-VOA1 staining, and found consistent results (Extended Figure 4).

To clarify this question, we are not calculating surface area but rather a maximal cross-sectional area. Briefly, this is accomplished by detailed analysis of each plaque, going through the high-resolution confocal stack for each spheroid and identifying the optical section showing the largest diameter. Using NIH/Image J we traced the boundaries of the spheroid at the maximal diameter slice and calculated the area using this software. We have now improved clarity in our methods section.

3. The authors ignore in their article that PLD3 is associated to AD in a loss of function scenario by the original study (Cruchaga et al). Thus it should be discussed why the authors think that the previous interpretation was wrong. Do the authors have evidence for upregulation of PLD3 in AD?

The reviewer is correct that the original study assumed a loss of function in the PLD3 gene to be associated with increased risk for AD. This was supported by measurements in AD patients showing lower levels of PLD3 gene expression (Figure 2 from the Cruchaga paper, PMID: 24336208). This same paper initially proposed that PLD3 mutations may lead to loss of function because they show a disrupted role in APP processing and subsequent amyloidosis. However, several subsequent papers contradicted this finding (PMID: 28128235).

Importantly, the precise function of PLD3 is unknown and there is ongoing controversy as to whether PLD3 is truly a phospholipase versus an exonuclease (PMID: 30111894). We do not think that there is currently any conclusive evidence one way or another as to the effect of PLD3 V232M variants on PLD3 function. What is clear from our study is that PLD3 is highly accumulated in spheroids, and that increasing PLD3 via overexpression worsens spheroid pathology while decreasing it improves this pathology.

We have also added new neuropathology data acquired from rare brains of patients with PLD3 V232M mutations. While limited to 4 postmortem brains that were available, it shows that this mutation is associated with a marked increase in the number of large “ring-like” vesicular structures within spheroids (Figure2). This is interesting because like with PLD3 overexpression and mouse aging, spheroids have a greater abundance of large ring like vesicular structures within (Figure 3).

Altogether, while our data is not conclusive with respect to the gain or loss of function question, it provides a strong case in favor of a gain of function, or a gain of toxic function associated with PLD3 accumulation or PLD3 variants. Future

research will be required for solving this question and this will likely require first discovering the precise physiological role of PLD3.

4. The authors speculate about a direct interaction between PLD3 and Abeta. This needs further clarification.

We would like to clarify that we are not proposing a direct interaction between PLD3 and A β . Rather we are showing the potential subcellular colocalization of PLD3 and A β and the uptake of A β into endosomes at PAAS. Combined with previous literature showing that A β can induce the formation of large endolysosomal vesicles (PMID: 28835279), similar to what we find with PLD3 overexpression, we are suggesting that there could be synergistic effects between these two molecules, which could potentiate their effect on the formation of aberrant endolysosomal vesicles and PAAS enlargement. The precise mechanisms of how this might operate are not known but could be the topic of future investigation. We have now extended our discussion on this topic.

Minor remarks

1. R175: ref 28 Fazarri et al provided evidence from in vivo experiments, the work encompassed crosses of PLD3 ko with APP knock in mice.

We have modified the text to correctly acknowledge the work from Fazarri et al.

2. Fig 1 d needs some more explanation. What is the green spot in the circle? What are the green cells depicted on top of the corpus callosum? I do not understand what happens: it seems that the cell bodies in the contralateral side are stimulated, but how is then t1 and t2 determined, i.e. how do you know that the signal in t1 is upstream of the spheroid and in t2 downstream?

We have now improved the clarity of this diagram and the associated figure legends to explain better the methodology.

In this particular experiment we cannot provide unambiguous anatomical evidence as to which ROI is upstream or downstream of the cell body because this would require highly challenging tracing experiments not feasible in vivo. However, we make the reasonable assumption that the action potential will arrive at the upstream segment sooner than the downstream segment. We thus calculated the absolute values of the calcium rise time difference between the two segments for individual trials and took the average from the trials for each axon. This allowed us to calculate a net conduction delay in axons with spheroids that was absent in segments of the same axon without spheroids or in normal axons.

3. Fig 1 e and e' : what is depicted in the Y-axis of the panels?

We have now clarified the Y axis better in the figure and figure legend , indicating that it describes the delta F/F of the intensity of Calcium traces.

4. Fig 1g AS is used as abbreviation in legend, do the authors mean PAAS? AS is axonal segment and also used as antisense in other parts of the MS.

Yes indeed we mean PAAS. We have checked throughout the text in this revision and all the abbreviations should be consistent now.

5. Fig 1g Is the number of axons measured not very small (n=10, 21 and 8 and are obtained from 14 mice, implying that <3 axons per mice were checked)

The reviewer is correct. This is a very challenging experiment making it difficult to obtain more axons per mouse. Briefly, in order to measure the local effect of PAAS-associated conduction defect, we had to isolate sparsely labeled axons in which both the spheroid and the up and downstream axon segments were clearly identifiable and parallel to the imaging plane. Importantly, these axons had to also be responsive to our contralateral electrode stimulation. Altogether the combination of these requirements reduced the probability of finding suitable axons for recording. However, while the N for each mouse is relatively low, the overall results we obtained are statistically robust and reproducible.

Furthermore, to supplement the low sample size of this difficult experiment, we also designed the experiment to look at population responses (Figure 1h-j), that yielded tens of axons imaged for each mouse. That experiment provided consistent results with the data obtained from individual axon imaging. We thus believe these two Calcium imaging experiments, in combination with our new voltage imaging experiments, complement each other and together support our conclusion of PAAS-induced conduction disruption.

6. Fig 2c: it is unclear how the panel has been generated. As the dots represent means of measurements, I do not understand what the coloured bars mean. From these data it seems that all PAAS have very similar sizes (60 μm^2)? It might be more meaningful if for each spot the mean \pm SD is given so that we get an understanding of the spreading in those data points

Each dot in the graph represented the average measurements from all the individual PAAS imaged in an individual mouse, with error bars showing the variance among different animals. We think this is appropriate since in these graphs we were trying to compare the group-level difference. Regarding the spread of individual PAAS sizes, we now have in Extended Data Figure 1 a distribution plot for all PAAS showing a log-normal distribution.

7. Fig 2k: the panel shows cathepsin d and APP staining. I suppose that the APP staining reflects here the spheroids, but then there is clearly no colocalisation with cathepsin D, or vice versa and how does this picture relate to the panels at the right?

To clarify, Cathepsin D and APP are indeed well colocalized when we image individual spheroids at high resolution (panels on the right). However, when viewing unzoomed images (left panel) the degree of colocalization is less apparent. This is especially true for cathepsin D, because of the very intense and thus highly saturated images of cathepsin D present within surrounding microglia, which makes the levels within spheroids appear lower. We have now clarified this in the figure legend which has now been relocated to Extended Data Figure 6.

8. Fig 3c and c': not indicated which area is zoomed in. I suppose that the GFP overexpression is coupled to GFP overexpression?

We thank the reviewer for pointing this out. Our initial images were actually different examples rather than zoomed versions. However, we now realize that this generates confusion and have replaced the panels for zoomed versions of the original low power views.

Referee #2 (Remarks to the Author):

This is a peer review for a new manuscript by Prof Grutzendler and colleagues wherein they describe the impact of dystrophic neurites on action potential conductance. They discovered that dystrophic neurites slow the conductance of action potentials and that this relationship is related to the size of dystrophic neurites, larger neurites producing greater slowing or even blocking conductance. They describe the presence of multivesicular bodies in dystrophic neurites, which they postulate are associated with larger size of the neurites. They report that phospholipase D3, a variant of which is reportedly a genetic risk factor for AD, is associated with these large multivesicular bodies. Overexpression of PLD3 via neuronally targeted expression driven by a viral vector resulted in expansion of dystrophic neurites while CRISPR-Cas9 mediated knock-down resulted in smaller dystrophic neurites. As with previous work from this group, the quality of the live-imaging microscopy is exceptional and creates a lot of visual appeal for the manuscript. Quantitative methodologies are used throughout the work and, while I am not a statistician, the statistical approaches seem reasonable. In a couple of instances specifically noted below, the experiments had too small of a sample size.

Figure 1: Establishing the electrophysiologic significance of dystrophic neurites has implications for connectivity and potentially for cognition, thus it is a priority research topic. Using live calcium imaging to visualize action potential conductance across dystrophic neurites is novel and from this data the authors conclude that dystrophic neurites act as capacitors to slow (or block) action potential conductance. Importantly, calcium transients are slower than action potential generation/conduction within axons and some calcium channels are low-threshold channels which may not

need action potentials to be activated. Consequently, while slowed action potential conductance across dystrophic neurites is a reasonable hypothesis, I am not sure the claim can be robustly supported using the calcium imaging method. Direct electrophysiological measurement or use of a voltage sensitive dye or protein may provide informative alternatives.

We the reviewer for this comment. In response, we have now performed in vivo voltage imaging of PAAS-associated conduction defects using the genetically encoded ASAP3 sensor (PMID: 31835034). To date, no group has demonstrated reliable in vivo voltage imaging in single axons due to low signal to noise ratio of these sensors. We were, however, successful at measuring antidromic axonal transmission by electrically stimulating the axon terminals while imaging the voltage sensor fluorescence in the soma (as previously described PMID: 31835034). Using this approach, first in wildtype mice, we could detect reliable action potentials in the soma in an all-or-none fashion, when stimulating the axon terminals above a certain current threshold. In contrast, in 5xFAD mice we observed much greater heterogeneity, and an overall requirement of greater currents to overcome conduction failure, consistent with the hypothesis of increased capacitance resulting from the presence of PAAS. Furthermore, when we used a high current to ensure transmission, we found that 5xFAD mice still showed a transmission delay for tens of milliseconds. We have added these new results to Figure 1. Together these new voltage imaging data provide more definitive support to our original conclusion, based on calcium imaging, that PAAS cause axonal conduction block and delay.

The modeling of the extent of axonal dystrophy associated with each plaque is of interest. I am not sure I agree with all of the assumptions the authors use in this analysis (notably, unlike the 5xFAD model, many plaques in human brain have no dystrophic neurites at all), but the calculations are probably useful to at least provide a sense of the scale of the process.

We completely agree with this comment. The intention of the modeling was to help us and readers to conceptualize the effect of axonal spheroids on action potential conduction. We did not expect the exact parameters used in our models to recapitulate the heterogenous and complex conditions in the human brain. On the second point made by the reviewer, we also reported plaques with no dystrophic neurites in a previous publication (PMID: 27196974). In the future, it would be very interesting to understand why neurons seem to be impervious to some types of plaques.

The conclusion that dystrophic neurites are a modifiable pathology rather than degenerative based on long-term live imaging in 5xFAD mice is somewhat overstated. The 5xFAD mouse does not model neurodegeneration or mature tau pathology. Dystrophic neurites in human tissue often contain tau aggregates and likely contain a different balance of organelles than the 5xFAD mouse – it remains quite plausible that the equivalent process in human brain is less reversible. I would moderate the interpretation of this result, but never-the-less, the long-term microscopy of these structures is an accomplishment of considerable interest.

We agree with the reviewer that human PAAS could potentially show different dynamics, and now toned down this statement by indicating in the discussion the potential differences with humans.

Figure 2: I am not convinced that the large LAMP1-positive vacuoles that are seen in the immunohistochemistry of dystrophic neurites represent the same type of organelle as that shown in the electron microscopy. The vacuoles on IHC seem quite a lot larger than the representative MVB on the EM. There are also other organelles adjacent to the MVB on the representative EM image that are of similar size. Conceptually, quite a few organelles could be candidates for a LAMP+ vacuole, including pathologically enlarged lysosomes or organelles in the later stages of the autophagic pathway. I also think it is important to verify the result in human tissue as there is some discrepancy in the descriptions in prior work of the types of organelles in dystrophic neurites between animal models and human neuropathological studies. We agree with the reviewer that the original EM image was not a good representation of the large Lamp1+ vesicles on IHC. And we also agree that at this point the evidence supporting the enlarged Lamp1-positive ring-like structures being multivesicular bodies remains correlative. We toned down the related language throughout the text and now refer to these as enlarged Lamp1-positive vesicles (now termed as ELPVs), acknowledging that multiple abnormalities along the endolysosomal and autolysosomal pathway could contribute to the enlargement of the spheroids. We also improved our EM data by providing examples of the diverse kinds of vesicle present within PAAS. Ultimately, the only way to

unambiguously link our confocal and EM images would be through correlated Confocal/TEM imaging, but we feel that this would be a very challenging experiment that while nice would not necessarily change our conclusions. We also agree that while there is already extensive EM literature describing dystrophic neurites in humans, there is room for much more work including potentially correlated light and EM imaging as well as 3D FIB/SEM. We have now added a commentary in the discussion about the potential for these future experiments, which we feel are beyond the scope of this paper.

I found the connection between the extent of axonal dystrophy pathology and cognition to be tenuous. The analysis of human tissue included no control subjects, 6 subjects with mild cognitive impairment at advanced ages and 12 subjects with AD over a range of ages. Aside from the categorical characterization, there was no information on their cognitive status. MCI, of course, represents a range of underlying processes (including in many cases normal aging) and it is possible that non-amyloid pathologies contributed. The use of APP as the marker of dystrophic neurites in this setting is questionable (fig2k shows the labeling with APP in human brain inconsistently marked dystrophic neurites). I would have preferred the authors utilized existing, larger-scale patient cohorts with neuropathological data to establish a rigorous association between dystrophic neurites with cognition.

We thank the reviewer for this comment. Numerous previous studies have already studied the correlation between plaque-associated dystrophy pathology and cognitive decline using postmortem brain tissue from AD patients. While these studies applied various labeling methods (APP, SMI, Tau), the general results are that dystrophic neurites correlated well with cognitive decline (PMID: 15184601, PMID: 9829809, PMID: 11483304). We have now cited these studies. As with many human studies we agree that there is heterogeneity, however, we used well characterized samples and patients derived from well-known Alzheimer centers. With regards to non-demented controls, they generally have very low plaque load and therefore it was very difficult to compare them rigorously with MCI and AD patients which have many more plaques. Overall, while the N= in our study are not huge, we feel that our data is statistically robust and is a useful addition to our study.

With regards to APP as the marker of dystrophic neurites, we have now added quantification with a second marker (endolysosomal v-ATPase subunit VOA1) that robustly labels PAAS in humans (Figure 2), showing results that are consistent with the ones with APP labeling (Extended Figure 4).

In addition, we have also obtained rare postmortem brains with PLD3 V232M mutations and have now shown that spheroids in these brains have more abundant enlarged endolysosomal vesicles within PAAS (as measured by imaging VOA1 labeling). This new data is included in Figure 2.

Figure 3: I do not find the evidence linking the staining pattern of PLD3 in dystrophic neurites to MVB convincing. PLD3 seems to be distributed throughout the dystrophic neurite, with some membrane associated and some not, as others have previously reported. The suggestion that PLD3 is linked to the biogenesis of MVBs is not supported by this data and does not consider the possibility of failed maturation or trafficking defects.

To clarify, the diffuse appearance throughout PAAS is mainly seen when using conventional confocal imaging. When we used expansion microscopy, it becomes obvious that PLD3 appears as a punctate label that sometimes colocalizes with the membranes of Lamp-1 positive vesicles while other times appeared to be within their lumen (Figure 3b). We had initially named these vesicles as multivesicular bodies based on some of their morphologies and previous publications showing PLD3 trafficking to the intraluminal vesicles of MVBs. However, as per the reviewer suggestion, we have now changed to a broader term that includes the possibility of other transitional and related endolysosomal and autophagic vesicles. We also clarified that the effect of PLD3 modulation may extend beyond MVBs to other inter-related endolysosomal organelles and as the reviewer states, it may relate to failed maturation. Although we favor less the defective trafficking hypothesis, we cannot formally rule it out with our data.

I did find the panels linking PLD3 expression to the size of dystrophic neurites compelling (panels C-H), although in prior figures the analysis was shown both per-plaque and per-mouse; that pattern should be continued. We don't really understand what factors are central to dystrophic neurite formation and growth, so this result is informative.

As requested, we have now added quantification by individual spheroids in Figures 3e', 4e', and Extended Figure 8b', 10c'.

The final panels in this figure (I-N) are under-developed and may be unnecessary. In a mouse model with non-physiologic levels of over-production of β -amyloid, it is not surprising that β -amyloid is present in neuronal lysosomes and certainly does not strongly support the notion that it was endocytosed. This experiment could have been performed in human tissue with a more convincing result. The use of Dynasore in the subsequent experiment is not discussed in the text or methods at all, there are no methods given for the slice culture, and the interpretation of the result is underdeveloped. In the final panels, the preparation of the β -amyloid for injection is not described, the number of replicates is small, no controls are used and there are a number of confounders and alternative explanations for these experiments which are not discussed.

We have now added details about all the methodologies and analyses as well as some potential limitations of this experiment (see methods).

With regards to the source of abeta within Lamp1 positive vesicles we had previously acknowledged in the discussion that we could not rule out endogenous production (given the known presence of APP and BACA1 within PAAS). However, and precisely because in humans there is generally no APP overexpression, we think that the endocytosis experiment is useful. It at least demonstrates the significant potential for endocytosis within PAAS as a possible source of Abeta, given the massive presence of ABeta in the surrounding amyloid deposits. We would rather keep this experiment as it currently is as we do think it adds useful insights.

Figure 4: Silencing PLD3 expression reduced the size of dystrophic neurites. This is a striking finding. Importantly, the demonstration that PLD3 knock-down in mouse neurons resulted in reduced size of dystrophic neurites set up the opportunity to causally demonstrate a link between dystrophic neurites and cognition through neurobehavioral testing. Cognitive data would more convincingly demonstrate that PLD3 is a reasonable drug target and would strengthen this work.

Cumulatively, I found the evidence that PLD3 regulates the size of dystrophic neurites convincing and potentially of considerable importance. Strengthening the link between PLD3 in dystrophic neurites and cognition would greatly enhance the study. The conclusion that vacuoles in dystrophic neurites are multivesicular bodies and that dystrophic neurites impair action potential conductance cannot be conclusively made from the available data, in my opinion.

We thank the reviewer for this important question. The same point was brought up by reviewer 3. To address this and other reviewer's concerns, we have now performed new experiments to examine the physiological impact of correcting PAAS pathology on neural circuit function (Figure 5). Specifically, we focused on basal forebrain projections to the cortex. We implemented in vivo calcium imaging of spontaneous activity in cortical neurons of awake mice. We previously treated these mice using CRISPR/Cas9 by viral injections into the basal forebrain with AAV vectors either encoding PLD3 sgRNA or control sgRNA. Consistent with previous reports we found aberrant hyperactivity and correlated activity patterns in the cortex of 5xFAD mice (PMID: 18802001). Importantly, we found that PLD3 knock-down in the basal forebrain neurons which project to the cortical neurons where we recorded calcium imaging, led to a marked improvement in the hyperactivity and correlated activity patterns of these neurons, bringing it close to normal levels in wildtype mice. These new results are consistent with the interpretation that a reduction in PAAS leads to a substantial improvement in cholinergic axonal conduction and neurotransmission. This is consistent with the known effect of cholinergic forebrain inputs, which exert an activating effect on layer II/III inhibitory GABAergic cortical interneurons, thereby reducing overall cortical aberrant excitation (PMID: 27657448).

We think this new data is exciting and an innovative and sensitive approach for demonstrating neural circuit effects of gene therapy treatments in AD-mice. While behavioral tests are possible, we do not favor their use in this case since multiple neural circuits might be involved in the execution of even a simple cognitive task. Thus, partially restoring such circuits may not lead to measurable behavioral improvements.

Referee #3 (Remarks to the Author):

The manuscript identifies plaque-associated axonal spheroids (PAAS) as contributors to action potential conduction delays and blockades in AD mice using calcium imaging of single axons in vivo. PAAS were stable over time and larger size was associated with increased conduction deficits. PAAS accumulated large multivesicular bodies (MVBs) and PLD3, a lysosomal protein. Mechanistically, overexpression of PLD3 led to accumulation of MVBs, PAAS enlargement and action potential conduction deficits, whereas PLD3 depletions reduced these deficits. Overall, the manuscript suggests that PLD3-dependent PAAS may contribute to network dysfunction in AD.

The manuscript addresses functionally for the first time and in a rigorous way one of the hallmarks of AD pathology, axonal dystrophic neurites, and identifies PLD3 as a potential mechanism of PAAS-induced action potential alterations. The experimental design is rigorous and elegant assessing single axon function with contralateral injections of tracers. Overall, the main conclusions are well supported by the experimental data, and results are properly discussed. I did not identify major deficiencies, but I have a few comments:

- 1) The authors repeatedly argue that PAAS contribute to “neural network dysfunction”. However, there is no assessment of neural network dysfunction by electrophysiological (e.g., EEG/LFP recordings) or behavioral approaches. GCaMP6f imaging before and after PAAS in single axons is not a measure of “neural network dysfunction”. There is only assessment at the axonal level and it seems too speculative to link these findings to network functions.
- 2) The overall conclusion that PLD3-dependent PAAS contribute to network dysfunction is weakly supported since no measures of network dysfunction are assessed in 5xFAD mice after PLD3 deletion.

We thank the reviewer for this important question. The same point was brought up by reviewer 2. To address this and other reviewer’s concerns, we have now performed new experiments to examine the physiological impact of correcting PAAS pathology on neural circuit function (Figure 5). Specifically, we focused on basal forebrain projections to the cortex. We implemented in vivo calcium imaging of spontaneous activity in cortical neurons of awake mice. We previously treated these mice using CRISPR/Cas9 by viral injections into the basal forebrain with AAV vectors either encoding PLD3 sgRNA or control sgRNA. Consistent with previous reports we found aberrant hyperactivity and correlated activity patterns in the cortex of 5xFAD mice (PMID: 18802001). Importantly, we found that PLD3 knock-down in the basal forebrain neurons which project to the cortical neurons where we recorded calcium imaging, led to a marked improvement in the hyperactivity and correlated activity patterns of these neurons, bringing it close to normal levels in wildtype mice. These new results are consistent with the interpretation that a reduction in PAAS leads to a substantial improvement in cholinergic axonal conduction and neurotransmission. This is consistent with the known effect of cholinergic forebrain inputs, which exert an activating effect on layer II/III inhibitory GABAergic cortical interneurons, thereby reducing overall cortical aberrant excitation (PMID: 27657448).

We think this new data is exciting and an innovative and sensitive approach for demonstrating neural circuit effects of gene therapy treatments in AD-mice. While behavioral tests are possible, we do not favor their use in this case since multiple neural circuits might be involved in the execution of even a simple cognitive task. Thus, partially restoring such circuits may not lead to measurable behavioral improvements.

- 3) Mouse PAAS are identified with a LAMP1-positive multivesicular bodies (MVBs) antibody, whereas human PAAS are labeled with APP or VOA1 (ATPase H transporting V0 subunit A1) antibodies, suggesting that human and mouse PAAS are not mechanistically equivalent. The authors also use the term “MVB-like structures” for human PAAS. Are mouse and human PAAS molecularly distinct?

The reviewer asked an interesting question regarding the differences of PAAS between mice and humans. While we are not claiming that all the features are identical, we do believe there are great mechanistic similarities. For example, previous studies examining the EM ultrastructure of PAAS in both species revealed very similar vesicular contents in

both species (PMID: 14119171, 8795633). Specifically, PAAS are filled with tightly packed vesicles including multivesicular bodies, lysosomes, autophagosomes, and vesicles at various intermediate stages (see diagrams in Extended data figure 5 and 12). In terms of antibodies to detect these structures, there are also many similarities but it is hard to compare given challenges due to different staining conditions, fixation protocols and antibody species. In general, many common markers exist that recognize proteins in both species, associated with endolysosomes, MVBs or autolysosomes in axonal spheroids such as PLD3, APP, LC3, CathepsinD and VOA1. Unfortunately, LAMP1 did not give a robust enough labeling in humans with high enough signal to background ratio where we can clearly recognize the boundaries of each spheroid for rigorous quantification. Thereby we did not use LAMP1 in humans. Instead, we used VOA1 which gives a very beautiful labeling in both species. While we agree that it would be nice to use the exact same markers in both species, it would be very time consuming to repeat these experiments and we do not think this fundamentally changes our conclusion given the strong colocalization of VOA1 and LAMP1 as seen in mice (see Extended Data Figure 6). However, we do recognize that there are likely differences between species that need to be taken into account in the future. In fact, we are currently in the process of doing an extensive proteomic study of spheroids in both species and in the future, we are hoping to learn about the similarities and differences. For example, a notable difference is the lack of labeling with phospho-Tau in mouse spheroids compared to those in humans. We have now added a comment in the discussion regarding the need for future detailed comparative characterization of the proteins and organelles within these axonal structures.

Minor:

1) The term “action potential spike time” is not accurate since calcium signal relates to trains of action potential during the 500-ms stimulation. Calcium rise time would be more appropriate for describing the data.

We thank the reviewer and have now corrected the terminology throughout the text. Importantly, we have also added new data in response to related comments by reviewer #1. We implemented in vivo voltage sensor imaging, and demonstrated good correlation with the data obtained by calcium imaging (Figure 1).

2) Inserts in calcium figures do not contain temporal scale so it is difficult to assess the time.

We have now added scale bars for time to the inserts.

3) Electrical stimulation is marked by an arrow giving the impression that is very brief, but these stimulations are very long for action potential dynamics (500 ms). Please indicate with a line the 500 ms of stimulation to properly assess the calcium data.

To clarify, we are not continuously stimulating for 500ms; rather we are giving a train of electrical pulses that are 2 ms ON and 18 ms OFF, over the 500ms interval. We have now labeled this on the graph, and made sure that the parameters are very clear in both figure legend and methods section.

4) Fig 4g, calcium signal is observed before the stimulation, why?

Thanks for this observation. We agree that it appears as if the signal came before stimulation. However, in the high-resolution inset, which was zoomed in from the exact same low zoom image it appears clear and shows that calcium change occurs after stimuli. We now changed the thickness of the arrow in the low resolution image which we think improves the appearance and clarity.

5) The term “plaque-associated axonal spheroids” needs some clarification, since PAAS are also observed far from plaques (e.g., extended figure 4).

To clarify, the spheroids seen in extended figures that were not associated with plaques, only occurred following PLD3 overexpression. In the setting of AD, we only see spheroids in close association with plaques, and thereby we used the term PAAS throughout the paper. In the case of PLD3 overexpression, for the sake of clarity, we only call them spheroids.

Reviewer Reports on the First Revision:

Referees' comments:

Referee #1 (Remarks to the Author):

Thank you for responding to my questions. I agree with these additional comments, but I believe that it is important to incorporate better the caveats and limitations of the study in the manuscript as well. It is important that the readers understand that some of the mechanisms and interpretations proposed remain speculative or correlative, as I said in my original review.

Referee #2 (Remarks to the Author):

The authors have provided a thoughtful and thorough response to my previous comments and the additional data is informative. I think the core results of the manuscript are exciting and I am enthusiastic to see this work published.

I continue to find the link to human cognition tenuous, but I understand the authors' preference to include it.

As I mentioned in the initial peer review, the quality of the microscopy overall is outstanding. This research group has a track record of innovative microscopy, particularly with in vivo imaging of AD pathologies. This ability to visualize pathologies longitudinally helps with both conceptualizing these processes and measuring the dynamics and relationships among pathologies.

I have some thoughts around the quantification for Fig 2e/f and 2g/h. I don't think these associations are best illustrated with a dichotomized outcome. Both pH and the cathepsin signal can and should be continuously measured and their association with the size of the dystrophic neurites reported. Additionally, for the pH experiments, I'm not sure the representative image strongly supports the quantified conclusion – visually my impression is that both large and small neurites have a quite a range of pHs. The suggestion that lysosomes and related organelles in dystrophic neurites fail to acidify is likely true based on a parallel, indirect lines of evidence, but it is a particularly important conclusion of this figure.

In figure 3, I have some reservations, in general, about expansion microscopy, as I suspect the expanded tissue is morphologically distorted and likely damaged by the expansion, but the traditional confocal imaging supports the conclusions adequately.

As noted in the prior set of comments, my major remaining reservation is the generally low number of biological replicates (animals) in each experiment (particularly re: figure 3). Especially when there are some signs of discrepancy with prior reports, ensuring adequate biological variability has been captured in the analyses is important. The results of figure 3 and extended figures 8/9 seem to conflict with the earlier report from Fazzari/de Strooper et al 2017. That study reported PLD3 KO increased neuronal lysosome size while the PLD3-silencing experiments in the current study do not appear to show similar changes and the overexpression experiments seem to produce endolysosomal enlargement (at least in dystrophic neurites). If the authors have data from analogous experiments in wild type mice, it would be of interest to include them as a counterpoint to the earlier results from Fazzari et al and to clarify if the discrepancy is contributed by the presence

of beta-amyloid or features of the microenvironment of the dystrophic neurite. More importantly, the group sizes are particularly small in the analyses surrounding lysosome size and it would increase my confidence in the conclusion to expand the analysis.

Referee #3 (Remarks to the Author):

The manuscript has been thoroughly revised and all my concerns have been properly addressed, particularly with the assessment of circuit and neuronal function (new Figure 5). Overall, the manuscript identifies plaque-associated axonal spheroids (PAAS) as key AD-related pathological change contributing to action potential conduction delays and blockades using calcium imaging of single axons in vivo. PAAS were identified in humans with AD and mice and stable over time, and its size was associated with increased action potential conduction deficits. PAAS accumulated large multivesicular bodies (MVBs) and PLD3, a lysosomal protein. Mechanistically, overexpression of PLD3 led to accumulation of MVBs, PAAS enlargement and action potential conduction deficits, whereas PLD3 depletions reduced these deficits. Overall, the manuscript demonstrates that PLD3-dependent PAAS contributes to action potential and network dysfunction in AD.

Excellent manuscript. I have no additional comments.

Author Rebuttals to First Revision:

Response to reviewer#1

Thank you for responding to my questions. I agree with these additional comments, but I believe that it is important to incorporate better the caveats and limitations of the study in the manuscript as well. It is important that the readers understand that some of the mechanisms and interpretations proposed remain speculative or correlative, as I said in my original review.

We appreciate this concern. We have gone through all the text and have tried to introduce new language or tone down the existing one to make sure there is a clear understanding that some of the findings are correlative. This is especially relevant for findings related to clinical-pathological correlations in humans.

Response to reviewer#2

The authors have provided a thoughtful and thorough response to my previous comments and the additional data is informative. I think the core results of the manuscript are exciting and I am enthusiastic to see this work published. As I mentioned in the initial peer review, the quality of the microscopy overall is outstanding. This research group has a track record of innovative microscopy, particularly with in vivo imaging of AD pathologies. This ability to visualize pathologies longitudinally helps with both conceptualizing these processes and measuring the dynamics and relationships among pathologies.

1) I continue to find the link to human cognition tenuous, but I understand the authors' preference to include it.

We appreciate the reviewer's understanding. Correlations of neuropathological features with human cognitive data are always challenging but we think they are important for the field, and it would be a missed opportunity not to report these findings. Similar to comments by reviewer 1, we have modified the language to describe the clinical pathological correlations to emphasize that these types of data always have limitations and should be interpreted with caution.

2) I have some thoughts around the quantification for Fig 2e/f and 2g/h. I don't think these associations are best illustrated with a dichotomized outcome. Both pH and the cathepsin signal can and should be continuously measured and their association with the size of the dystrophic neurites reported.

As requested, we have now provided the same data displayed as a continuous quantification showing hundreds of individual spheroids analyzed for both pH and cathepsin, see new Extended Figures 6f and g demonstrating a clear correlation between pH (green/red ratio), cathepsin and size. However, we would like to put this in in Extended Figure 6 while keeping the original binarized quantification in the main figures because we think this is a much easier way to visualize and understand the data for the general reader.

3) Additionally, for the pH experiments, I'm not sure the representative image strongly supports the quantified conclusion – visually my impression is that both large and small neurites have a quite a range of pHs. The suggestion that lysosomes and related organelles in dystrophic neurites fail to acidify is likely true based on a parallel, indirect lines of evidence, but it is a particularly important conclusion of this figure.

We appreciate the reviewer's concern. However, simple "eyeballing" of the spheroids in our figure is an inaccurate way to make any conclusions given the heterogeneity of spheroids, the different degrees of viral transfection of the pH reporter in each axon and the complexity of our quantifications. The analyses we did relies on precise ratiometric fluorescence intensity quantifications between green and red fluorescence using customized computer macros applied to hundreds of individual spheroids coupled with simultaneous measurement of spheroid area (now shown in extended data Fig. 6). We are thus very confident that our data is highly quantitative, robust and reproducible.

4) In figure 3, I have some reservations, in general, about expansion microscopy, as I suspect the expanded tissue is morphologically distorted and likely damaged by the expansion, but the traditional confocal imaging supports the conclusions adequately.

We are happy that the confocal images are acceptable to the reviewer. However, we would like to keep the expansion images in the paper. This technique has been shown in rigorous quantifications from the Boyden lab at MIT and others to expand the tissue isometrically without causing meaningful tissue distortions. The additional resolution afforded by this technique reveals useful details about the subcellular location of PLD3 within LAMP1 positive vesicles.

5) The results of figure 3 and extended figures 8/9 seem to conflict with the earlier report from Fazzari/de Strooper et al 2017. That study reported PLD3 KO increased neuronal lysosome size while the PLD3-silencing experiments in the current study do not appear to show similar changes and the overexpression experiments seem to produce endolysosomal enlargement (at least in dystrophic neurites).

We appreciate the concern of the reviewer regarding the difference between our results and that of Fazzari/De Strooper lab (Nature 2017). In addition to this paper there is also another relevant paper by Gonzalez et al (Cell Reports PMID29386126) contrasting with Fazzari et al, that shows no alterations in endolysosomal size after PLD3 deletion in vitro in Hela cells. We would like to clarify that we do not think these results necessarily contradict each other given their completely different experimental settings and even definitions of what is being quantified. We have now added a few lines in the discussion to address the complexity of the effects of PLD3 manipulations under different experimental conditions that include different ages, amyloidosis, cell body quantifications versus axons, etc.

The following are more detailed explanations of the potential reasons for the differences in results:

1) Fazzari et al. focused exclusively on quantifying classical lysosomes, while we quantified a broader population of aberrantly enlarged lamp1-positive vesicles, which as shown in our EM likely includes late endosomes, lysosomes, endolysosomes, amphisomes, and autolysosomes which are highly enriched within axonal spheroids. 2) Fazzari et al. focused exclusively on classical lysosomes visualized at neuronal cell bodies while our study focused on plaque associated axonal spheroids rather than cell bodies. This is a critical difference given that it is well known that there are major differences in lysosomes at cell bodies of neurons versus those in distant axons that undergo stepwise maturation and fusions as they are retrogradely trafficked to the cell body. 3) Fazzari et al. used 1-month-old mice, while our study was performed in 5- or 10-month-old AD-like mice, so both aging and amyloid context on top of our axonal focus are major differences. 4) In fact, our own data overexpressing PLD3 in wild type mice shows that

Lamp1 vesicles show no obvious changes at cell bodies while those in axons can be enlarged and associated with small spheroids even in the absence of amyloid pathology (Extended Figure 8). Of course, when the experiment is done in AD-like mice there is a dramatic worsening of LAMP1 vesicle enlargement in axons with formation of plaque-associated spheroids. Notice however, that enlarged LAMP1 vesicles in cell bodies are still not observed even in AD-like mice (Extended Figure 8). This strongly demonstrates the difference in endolysosomal maturation and dynamics in axons versus cell bodies, highlighting the susceptibility of long axons to pathology, something that Fazzari et al did not explore. 5) Fazzari et al. used a constitutive PLD3 global knockout mouse while we used viral mediated CRISPR/Cas9 deletion or overexpression only in neurons of adult mice. This difference is significant as gene deletion from birth could lead to compensatory changes in endolysosomal biogenesis, not seen in adult KO. Furthermore, using a global KO may introduce various unknowns regarding cell-cell interactions that are absent when targeting only a single cell type like in our case.

Altogether, these 3 studies demonstrate that there is a significant degree of complexity with regards to PLD3's role in lysosome biogenesis that will require further investigation in the future. Our data however clearly shows that LAMP1 positive vesicles are highly enlarged in PLD3 overexpression and reduced in size in PLD3 KO, when we analyze axons around plaques. This is not the case when we analyze LAMP1 vesicles at the cell body level like it was done by Fazzari et al.

6) As noted in the prior set of comments, my major remaining reservation is the generally low number of biological replicates (animals) in each experiment (particularly re: figure 3). Especially when there are some signs of discrepancy with prior reports, ensuring adequate biological variability has been captured in the analyses is important.

As noted above there are clear and parsimonious explanations for the differences in the results from the various studies. It is highly unlikely that this is explained by a low N# and that further data acquisition will change our results. As shown in all quantifications in figure 3 the data comparing control GFP and PLD3 overexpression are highly non overlapping and statistically highly significant. Notice for example the massively enlarged vesicles in images 3f and h and the overall quantification showing nearly a 300% difference in the size of spheroids (Figure 3e) and 100% difference for the size of LAMP1 vesicles, with completely non overlapping data and a p value of 0.007. These quantifications involved individual measurements in hundreds of spheroids. We do not think there is a justification for doing such complicated experiments again, just because our data does not fit exactly that of Fazzari et al, which was done under completely different experimental conditions.

Response to reviewer # 3

The manuscript has been thoroughly revised and all my concerns have been properly addressed, particularly with the assessment of circuit and neuronal function (new Figure 5). Overall, the manuscript identifies plaque-associated axonal spheroids (PAAS) as key AD-related pathological change contributing to action potential conduction delays and blockades using calcium imaging of single axons in vivo. PAAS were identified in humans with AD and mice and stable over time, and its size was associated with increased action potential conduction deficits. PAAS accumulated large multivesicular bodies (MVBs)

and PLD3, a lysosomal protein. Mechanistically, overexpression of PLD3 led to accumulation of MVBs, PAAS enlargement and action potential conduction deficits, whereas PLD3 depletions reduced these deficits. Overall, the manuscript demonstrates that PLD3-dependent PAAS contributes to action potential and network dysfunction in AD.

Excellent manuscript. I have no additional comments.